# Regression with Cost-based Rejection

**Xin Cheng**[1]   **Yuzhou Cao**[2]   **Haobo Wang**[3]   **Hongxin Wei**[4]   **Bo An**[2]   **Lei Feng**[2*]

[1]College of Computer Science, Chongqing University, China
[2]School of Computer Science and Engineering, Nanyang Technological University, Singapore
[3]School of Software Technology, Zhejiang University, China
[4]Department of Statistics and Data Science, Southern University of Science and Technology, China
xincheng9215@gmail.com, yuzhou002@e.ntu.edu.sg, wanghaobo@zju.edu.cn
weihx@sustech.edu.cn, boan@ntu.edu.sg, lfengqaq@gmail.com

## Abstract

Learning with rejection is an important framework that can refrain from making predictions to avoid critical mispredictions by balancing between prediction and rejection. Previous studies on cost-based rejection only focused on the classification setting, which cannot handle the continuous and infinite target space in the regression setting. In this paper, we investigate a novel regression problem called regression with cost-based rejection, where the model can reject to make predictions on some examples given certain rejection costs. To solve this problem, we first formulate the expected risk for this problem and then derive the Bayes optimal solution, which shows that the optimal model should reject to make predictions on the examples whose variance is larger than the rejection cost when the mean squared error is used as the evaluation metric. Furthermore, we propose to train the model by a surrogate loss function that considers rejection as binary classification and we provide conditions for the model consistency, which implies that the Bayes optimal solution can be recovered by our proposed surrogate loss. Extensive experiments demonstrate the effectiveness of our proposed method.

## 1   Introduction

In machine learning, the learned model from training data is expected to make predictions on unknown test data as accurately as possible. However, it would be unreasonable for the learned model to make predictions on all the test instances, as there may exist some difficult instances that the learned model cannot give an accurate prediction. Incorrect predictions can cause severe consequences and even can be life-threatening, especially in risk-sensitive applications [5, 19, 36, 11], such as healthcare management, autonomous driving, and product inspection. Therefore, the *learning with rejection* (LwR) framework [10, 11, 36, 27, 31, 17, 8, 6, 38] was extensively investigated, which aims to provide a reject option to not make a prediction in order to prevent critical false predictions. In this case, the LwR model can be learned by balancing the rejection and the prediction.

So far, most of the existing studies on LwR have focused on the classification setting, i.e., *classification with rejection* [10, 3, 44, 45, 14, 19, 37, 17, 8, 6]. A well-known framework for classification with rejection that has been studied extensively is called the cost-based framework, i.e., *classification with cost-based rejection* [11, 12, 18, 39, 16, 36, 8, 6]. In the classification with cost-based rejection setting, there is a pre-determined rejection cost $c$ for each instance, which must be smaller than the classification error $1$. A typical approach for classification with cost-based rejection is the *confidence-based approach* [22, 3, 45, 39, 40, 8]. The main idea is to use the real-valued output of the classifier as the confidence score and decide whether to reject the prediction based on the

---

[*]Corresponding author: Lei Feng.

confidence score and the given rejection cost $c$. Another effective approach is *classifier-rejector approach* [11, 12, 36], which simultaneously trains a classifier and a rejector, and this approach achieves state-of-the-art performance in binary classification [36].

Despite many previous studies on LwR, they only focused on the classification setting, which cannot handle the continuous and infinite target space in the regression setting. In many real-world scenarios, regression tasks with continuous real-valued targets can be commonly encountered. However, even state-of-the-art regression models may make incorrect predictions, and blindly trusting the model results may lead to critical consequences, especially in risk-sensitive applications [7, 4, 28, 23, 48]. Therefore, it is necessary to consider adding a rejection option for the regression problem to not make predictions in order to avoid critical mispredictions. To this end, many studies have been conducted on *regression with rejection*. A widely studied framework in regression with rejection is called *selective regression* [26, 44, 20, 46, 27, 41] that trains a regression model with a reject option given a fixed reject rate of predictions. However, this selective regression setting fails to consider the cost-based rejection scenario where a certain cost could be incurred if the model chooses to refrain from making a prediction for a certain instance.

In this paper, we provide the first attempt to investigate a novel regression setting called *regression with cost-based rejection* (RcR), where the model could reject to make predictions on some instances at certain costs to avoid critical mispredictions. To solve the RcR problem, we first formulate the expected risk and then derive the Bayes optimal solution, which shows that the optimal model should reject to make predictions on the examples whose variance is larger than the rejection cost when the popular mean squared error is used as the regression loss. However, it is difficult to directly optimize the expected risk to derive the optimal solution, since the variance of the instances cannot be easily accessed. Therefore, we propose a surrogate loss function to train the model that considers the rejection behavior as a binary classification and we provide theoretical analyses to show that the Bayes optimal solution can be recovered by minimizing our surrogate loss under mild conditions. Our main contributions can be summarized as follows:

- We propose a surrogate loss function considering rejection as binary classification and we demonstrate the *regressor-consistency* and the *rejector-calibration* when the binary classification loss function is classification-calibrated and is always greater than 0.

- We also provide a relaxed condition that allows the classification-calibrated binary classification loss to be non-negative. In the relaxed condition, the regressor-consistency can still be satisfied for correctly accepted instances.

- We derive a regret transfer bound and an estimation error bound for our proposed method, and extensive experiments demonstrate the effectiveness of our method.

## 2 Preliminaries

In this section, we introduce preliminary knowledge of ordinary regression, classification with rejection, and selective regression.

### 2.1 Ordinary Regression

For the ordinary regression problem, let the feature space be $\mathcal{X} \in \mathbb{R}^d$ and the label space be $\mathcal{Y} \in \mathbb{R}$. Let us denote by $(\boldsymbol{x}, y)$ an example including an instance $x$ and a real-valued label $y$. Each example $(\boldsymbol{x}, y) \in \mathcal{X} \times \mathcal{Y}$ is assumed to be independently sampled from an unknown data distribution with probability density $p(\boldsymbol{x}, y)$. For the regression task, we aim to learn a regression model $h : \mathcal{X} \mapsto \mathbb{R}$ that minimizes the following expected risk:

$$R(L) = \mathbb{E}_{p(\boldsymbol{x}, y)}[L(h(\boldsymbol{x}), y)], \tag{1}$$

where $\mathbb{E}_{p(\boldsymbol{x}, y)}$ denotes the expectation over the data distribution $p(\boldsymbol{x}, y)$ and $L : \mathbb{R} \times \mathbb{R} \mapsto \mathbb{R}_+$ is a conventional loss function (such as mean squared error and mean absolute error) for regression, which measures how well a model estimates a given real-valued label.

### 2.2 Classification with Rejection

A widely studied framework in classification with rejection is the cost-based framework [10, 36, 8, 6] that aims to train a classifier $f : \mathcal{X} \to \mathcal{Z}^{\circledR}$ that can reject to make a prediction, where ® denotes the

reject option. The evaluation metric of this task is the zero-one-c loss $\ell_{01c}$ defined as follows:

$$\ell_{01c}(f(\boldsymbol{x}), z) = \begin{cases} c(\boldsymbol{x}), & f(\boldsymbol{x}) = \circledR, \\ \ell_{01}(f(\boldsymbol{x}, z), & \text{otherwise,} \end{cases} \tag{2}$$

where $c(\boldsymbol{x})$ is the rejection cost associated with $\boldsymbol{x}$. Then, the expected risk with $\ell_{01c}$ can be represented as follows:

$$R_{01c}(f) = \mathbb{E}_{p(\boldsymbol{x},y)}[\ell_{01c}(f(\boldsymbol{x}), y)], \tag{3}$$

The optimal solution for classification with rejection $f^\star = \operatorname{argmin}_{f \in \mathcal{F}} R_{01c}(f)$ given by Chow's rule [10] can be expressed as follows:

**Definition 1.** *(Chow's Rule [10]) A classifier $f : \mathcal{X} \to \mathcal{Z}^\circledR$ is the optimal solution of expected risk (3) if and only if the following conditions are almost satisfied:*

$$f(\boldsymbol{x}) = \begin{cases} \circledR, & \max_z \eta_z(\boldsymbol{x}) \leq 1 - c(\boldsymbol{x}), \\ \operatorname{argmax}_z \eta_z(\boldsymbol{x}), & \text{otherwise,} \end{cases} \tag{4}$$

where $\eta_z(\boldsymbol{x}) = p(z|\boldsymbol{x})$. Chow's rule shows that classification with rejection can be solved when $\boldsymbol{\eta}(\boldsymbol{x})$ is known. However, the estimation of the probability is difficult especially when using deep neural networks [24].

## 2.3 Selective Regression

In selective regression, for a given instance $\boldsymbol{x}$, the selective model can choose to make a prediction for it or reject to make a prediction without costs. Formally, the selective model is a pair $(h, r)$ where $h : \mathcal{X} \to \mathcal{R}$ is a regression prediction model and $r : \mathcal{X} \to \mathcal{R}$ is a selection model, which serves as the rejection rule as follows,

$$(h, r)(\boldsymbol{x}) = \begin{cases} h(\boldsymbol{x}), & r(\boldsymbol{x}) > 0, \\ \circledR, & \text{otherwise.} \end{cases} \tag{5}$$

Let us denote by $\epsilon$ the expected rejection rate and denote by $\phi(r) = \mathbb{E}_{p(\boldsymbol{x},y)}\mathbb{I}[r(\boldsymbol{x}) > 0]$ the coverage of selective regression. The purpose of selective regression is to derive a pair $(h, r)$ such that the risk $R(h, r) = \mathbb{E}_{p(\boldsymbol{x},y)}[L(h(\boldsymbol{x}), y)\mathbb{I}[r(\boldsymbol{x}) > 0]]$ is minimized with coverage $1 - \phi(r) < \epsilon$, where $L(h(\boldsymbol{x}), y)$ is a conventional regression loss function. However, the selective regression setting fails to consider the cost-based rejection scenario but fixes the rejection rate $\epsilon$. In many real-world scenarios, rejection with costs is more common, and the cost $c(\boldsymbol{x})$ is easier to provide compared with the rejection rate $\epsilon$.

## 3 Regression with Cost-based Rejection

Let $\mathcal{X} \in \mathbb{R}^d$ be the $d$-dimensional feature space and $\mathcal{Y} \in \mathbb{R}$ be the label space. Suppose the training set is denoted by $\mathcal{D} = \{(\boldsymbol{x}_i, y_i)\}_{i=1}^n$, and each training example $(\boldsymbol{x}_i, y_i) \in \mathcal{X} \times \mathcal{Y}$ is assumed to be sampled from an unknown data distribution with probability density $p(\boldsymbol{x}, y)$. In RcR setting, for a given instance $\boldsymbol{x}$, the learner has the option $\circledR$ to reject making a prediction or to make a regression prediction. If the learner rejects an instance, the cost is a non-negative loss $c(\boldsymbol{x})$. The goal of RcR is to induce a pair $(h, r)$ where $h : \mathcal{X} \mapsto \mathbb{R}$ is a regressor to predict the accepted instance and $r : \mathcal{X} \mapsto \mathbb{R}$ is a rejector to determine whether to reject an instance or not. The evaluation metric of this task is the following loss function $\mathcal{L}(h, r, c, \boldsymbol{x}, y)$:

$$\mathcal{L}(h, r, c, \boldsymbol{x}, y) = \begin{cases} L(h(\boldsymbol{x}), y), & r(\boldsymbol{x}) > 0, \\ c(\boldsymbol{x}), & \text{otherwise,} \end{cases} \tag{6}$$

where $L(h(\boldsymbol{x}), y)$ is a conventional regression loss function (e.g., mean squared error).

In what follows, we will present a Bayes optimal solution to the RcR problem and provide a surrogate loss function to train the regressor-rejector.

### 3.1 Bayes Optimal Solution

In this paper, we only discuss the case where the loss function $L(h(\boldsymbol{x}), y)$ is the mean squared error (MSE), which is the most widely used regression loss function. The expected risk of $\mathcal{L}(h, r, c, \boldsymbol{x}, y)$ over the data distribution can be represented as follows:

$$R_{\mathrm{RcR}}(h, r) = \mathbb{E}_{p(\boldsymbol{x}, y)}[\mathcal{L}(h, r, c, \boldsymbol{x}, y)]. \tag{7}$$

Let us denote by $(h^{\star}, r^{\star}) = \operatorname{argmin}_{(h, r)} R_{\mathrm{RcR}}(h, r)$ the optimal pair of the expected risk $R_{\mathrm{RcR}}$ and we use $\mathbb{E}_{p(y|\boldsymbol{x})}[y] = \int_{\mathcal{Y}} p(y|\boldsymbol{x}) y \mathrm{d}y$ and $\mathbb{D}_{p(y|\boldsymbol{x})}[y] = \int_{\mathcal{Y}} p(y|\boldsymbol{x})(y - \mathbb{E}_{p(y|\boldsymbol{x})}[y])^2 \mathrm{d}y$ denote the expectation and variance of $y$ over the distribution $p(y|\boldsymbol{x})$. For a given cost function $c(\boldsymbol{x})$, we have the following theorem:

**Theorem 2.** *For a given instance $\boldsymbol{x}$, a non-negative cost $c(\boldsymbol{x})$ and the Bayes optimal pair $(h^{\star}, r^{\star})$ of the risk $R_{\mathrm{RcR}}$, the following equality holds:*

$$\begin{cases} h^{\star}(\boldsymbol{x}) = \mathbb{E}_{p(y|\boldsymbol{x})}[y], \\ r^{\star}(\boldsymbol{x}) = \mathbb{I}\left(c(\boldsymbol{x}) - \mathbb{D}_{p(y|\boldsymbol{x})}[y]\right). \end{cases} \tag{8}$$

The proof of Theorem 2 is provided in Appendix A. It is worth noting that our derived Bayes optimal solution in Theorem 2 can be considered as a generalized version of Proposition 2.1 in Zaoui et al. [46], with an instance-dependent cost function. Theorem 2 shows the expected optimal pair $(h^{\star}, r^{\star})$ of risk $R_{\mathrm{RcR}}$ where the rejector $r^{\star}$ should reject making a prediction if the variance of the distribution of labels $y$ associated with $x$ is so large that it exceeds a given rejection cost $c(\boldsymbol{x})$. This is intuitive and easy to understand. Unfortunately the probability density function $p(y|\boldsymbol{x})$ is usually unknown, meaning that obtaining the variance $\mathbb{D}_{p(y|\boldsymbol{x})}[y]$ and expectation $\mathbb{E}_{p(y|\boldsymbol{x})}[y]$ is difficult or even impossible. Many previous studies adopted specific assumptions to avoid directly estimating the variance and the expectation (e.g., homoscedasticity [26, 43, 42] and heteroscedasticity [29, 30, 9, 32]), while all of them have certain constraints. Therefore, the key challenge of RcR is how to learn the optimal solution $(h^{\star}, r^{\star})$ without the expectation and the variance.

### 3.2 Surrogate Loss Function of Training Regressor-Rejector

From Theorem 2, we know how the optimal pair $(h^{\star}, r^{\star})$ makes rejection and prediction for an unknown instance, but since the expectation and the variance are difficult to obtain, we cannot directly derive the optimal regressor and rejector. Let us reconsider the RcR loss function $\mathcal{L}(h, r, c, \boldsymbol{x}, y)$ by the following equation:

$$\mathcal{L}(h, r, c, \boldsymbol{x}, y) = (h(\boldsymbol{x}) - y)^2 \mathbb{I}[r(\boldsymbol{x}) > 0] + c(\boldsymbol{x}) \mathbb{I}[r(\boldsymbol{x}) \le 0], \tag{9}$$

where $\mathbb{I}[\cdot]$ denotes the indicator function. We cannot directly derive a regressor $h$ and a rejector $r$ by the above loss since the loss function contains non-convex and discontinuous parts $\mathbb{I}[r(\boldsymbol{x}) > 0]$ and $\mathbb{I}[r(\boldsymbol{x}) \le 0]$. In order to efficiently optimize the target loss, using surrogate loss is preferred. It is worth noting that the behavior of the rejector is similar to binary classification due to the only two options, rejection and acceptance. We may consider it directly as a binary classification $\{+1, -1\}$, where $+1$ means acceptance and $-1$ means rejection. Then we have the following surrogate loss function:

$$\psi(h, r, c, \boldsymbol{x}, y) = (h(\boldsymbol{x}) - y)^2 \ell(r(\boldsymbol{x}), -1) + c(\boldsymbol{x}) \ell(r(\boldsymbol{x}), +1), \tag{10}$$

where $\ell(\cdot)$ is an arbitrary binary classification loss function such as hinge loss. Then the expected risk with our surrogate loss $\psi$ can be represented as follows:

$$R_{\mathrm{RcR}}^{\psi}(h, r) = \mathbb{E}_{p(\boldsymbol{x}, y)}[\psi(h, r, c, \boldsymbol{x}, y)]. \tag{11}$$

The intuition behind this is that when the squared error is less than the given cost, we expect its weight $\ell(r(\boldsymbol{x}), -1)$ to be larger, i.e., $\ell(r(\boldsymbol{x}), +1)$ to be smaller. However, not every binary classification loss is theoretically grounded.

## 4 Theoretical Analysis

In this section, we first introduce the definitions of regressor-consistency and rejector-calibration. Then, we show the condition that our method can result in the Bayes optimal solution. Furthermore, we provide a relaxed condition that the regressor-consistency is only satisfied for correctly accepted instances. Finally, We derive a regret transfer bound and an estimation error bound for our method.

## 4.1 Regressor-Consistency and Rejector-Calibration

The rejector-calibration we are talking about here is related to the notion of classification calibration [1, 47, 15]. The notion of classification calibration of surrogate losses is defined as the minimum requirement to ensure that a risk-minimizing classifier becomes the Bayes optimal classifier, which is a pointwise version of consistency. The definition of rejector-calibration is given below.

**Definition 3.** *(Rejector-calibration) We say a rejector $r : \mathcal{X} \to \mathbb{R}$ is calibrated if $r$ always makes the same decisions as the Bayes optimal rejector $r^\star$ in Theorem 2, i.e., $\mathrm{sign}(r(\boldsymbol{x})) = \mathrm{sign}(r^\star(\boldsymbol{x}))$ for all $\boldsymbol{x} \in \mathcal{X}$ such that $r^\star(\boldsymbol{x}) \neq 0$.*

The definition of rejector-calibration indicates that we do not need to obtain the exact optimal rejector due to the difficulty of obtaining the variance, therefore, we just need to ensure that our rejector makes the same decisions as the optimal rejector in Theorem 2.

On the other hand, we define the regressor-consistency as follows.

**Definition 4.** *(Regressor-consistency) We say a regressor $h : \mathcal{X} \to \mathbb{R}$ is consistent if $h$ is equivalent to the Bayes optimal regressor $h^\star$ in Theorem 2, i.e., $h = h^\star$.*

Then we demonstrate that our method is regressor-consistent by the following theorem:

**Theorem 5.** *Suppose the binary loss $\ell$ is always larger than 0. For a given non-negative cost function $c(\boldsymbol{x})$, for any fixed rejector $r$, the optimal regressor $h_\psi^\star = \mathrm{argmin}_{h \in \mathcal{H}} R_{\mathrm{RcR}}^\psi(h, r)$ is equivalent to the Bayes optimal regressor $h^\star$.*

The proof of Theorem 5 is provided in Appendix B.1. Theorem 5 shows that the regressor $h_\psi^\star$ learned from our method can be equivalent to the Bayes optimal regressor $h^\star$.

It is worth noting that there is a special case $\ell(r(\boldsymbol{x}), -1) = 0$, where the regressor actually ignores the instance $\boldsymbol{x}$. Here we show a relaxed condition for regressor-consistency by the following theorem:

**Theorem 6.** *Suppose the binary loss function $\ell$ is classification-calibrated and is always non-negative. Given a non-negative cost function $c(\boldsymbol{x})$, for any fixed rejector $r$ and for the optimal regressor $h_\psi^\star = \mathrm{argmin}_{h \in \mathcal{H}} R_{\mathrm{RcR}}^\psi(h, r)$, the regressor-consistency can be only satisfied for correctly accepted instances (i.e., $\forall \boldsymbol{x} \in \mathcal{X}, r^\star(\boldsymbol{x}) > 0$).*

The proof of Theorem 6 is provided in Appendix B.2. Theorem 6 gives a relaxed condition of consistency, where the regressor-consistency is still satisfied for correctly accepted instances.

Then, we demonstrate that our method is rejector-calibrated by the following theorem:

**Theorem 7.** *Suppose the binary loss $\ell$ is classification-calibrated and is always larger than 0. For a given non-negative cost function $c(\boldsymbol{x})$, the optimal rejector $r_\psi^\star = \mathrm{argmin}_{r \in \mathcal{R}} R_{\mathrm{RcR}}^\psi(h_\psi^\star, r)$ is calibrated (i.e., $\mathrm{sign}(r_\psi^\star(\boldsymbol{x})) = \mathrm{sign}(r^\star(\boldsymbol{x}))$), where $r^\star$ is the Bayes optimal rejector.*

The proof of Theorem 7 is provided in Appendix B.3. Theorem 7 shows that our method is rejector-calibrated if the used binary loss $\ell$ is classification-calibrated.

## 4.2 Regret Transfer Bound and Estimation Error Bound

In the previous section, we have given the Bayes consistency analysis of our method, i.e., if the minimizer of our proposed risk can be the optimal one in Theorem 2. However, such a result does not guarantee the performance of models that are close to but not the minimizer of the risk $R_{\mathrm{RcR}}^\psi$, which occurs commonly since we usually minimize the empirical risk in practice. We provide a theoretical guarantee for such cases by showing the following regret transfer bound:

**Theorem 8.** *Suppose that $\forall \boldsymbol{x} \in \mathcal{X}, \forall y \in \mathcal{Y}, \mathbb{E}_{p(y|\boldsymbol{x})}[(h(\boldsymbol{x}) - y)^2] \leq M$ and $c(\boldsymbol{x}) \leq C$ hold almost surely:*

$$R_{\mathrm{RcR}}(h, r) - R_{\mathrm{RcR}}^* \leq \xi(R_{\mathrm{RcR}}^\psi(h, r) - R_{\mathrm{RcR}}^{\psi*})),$$

*This regret transfer bound holds for widely used binary losses, e.g., when $\ell$ is the sigmoid loss or the hinge loss, $\xi(u) = |u|$. When $\ell$ is the logistic loss or the square loss, $\xi(u) = \min\left\{2u, 2\sqrt{(M+C)u}\right\}$.*

The proof of Theorem 8 is provided in Appendix C.1. This theorem guarantees that even if the obtained $(h, r)$ is not exactly the minimizer of $R_{\mathrm{RcR}}^{\psi}$, we can also expect them to achieve good performance as long as they have a low risk $R_{\mathrm{RcR}}^{\psi}$. Then we can further get the following estimation error bound:

**Theorem 9.** *Suppose the binary loss is upper bounded by $M_1 > 0$ and $\rho$-Lipschitz continuous, $|h|$, $c(\boldsymbol{x})$, and $|y|$ are bounded by $M_2 > 0$. Given the empirical risk minimizers $\hat{h}$ and $\hat{r}$, the following bound holds with probability at least $1 - \delta$:*

$$R_{\mathrm{RcR}}^{\psi}(\hat{h}, \hat{r}) - R_{\mathrm{RcR}}^{\psi*} \leq 2\sqrt{2}L_1(\mathfrak{R}_n(\mathcal{H}) + \mathfrak{R}_n(\mathcal{R})) + C_1\sqrt{\frac{2\log(2/\delta)}{n}},$$

*where $C_1 = (4M_2^2 + M_2)M_1$, $L_1 = \sqrt{(4M_1^2\rho + M_1\rho)^2 + 16M_1^4M_2^2}$, $n$ is the i.i.d. sample size, and $\mathfrak{R}_n$ is the Rademacher complexity [2].*

The proof of Theorem 9 is provided in Appendix C.2. Given the fact that the Rademacher complexity usually decays at the rate of $\mathcal{O}(1/\sqrt{n})$, we can finally conclude that the performance of our model can approximate its optimal performance with the increasing size of the training set. In the following sections, we will demonstrate the effectiveness of our approach through experiments.

## 5    Experiments

In this section, we show the experimental results when our method is equipped with various binary classification losses and is compared with selective regression methods. In addition to the evaluation metrics commonly used for LwR, we propose additional evaluation metrics. Details of the experiment and the complete experiment can be found in Appendix D.

### 5.1    Implementation Details

When using deep neural networks as the model and using the gradient descent optimization, we consider a possible scenario where the regressor $h$ predicts any instance $\boldsymbol{x}$ with such a large error that $\ell(h(\boldsymbol{x}), y) >> c(\boldsymbol{x})$. In this case the rejector $r$ expects to reject all instances to make the empirical risk minimal. However, when the rejector $r$ converges quickly to reject all train instances, i.e., $\ell(r(\boldsymbol{x}), -1) \to 0$ for all train instances, the surrogate loss $\psi$ will be constant equal to $c(\boldsymbol{x})\ell(r(\boldsymbol{x}), +1)$. At that point the gradient of the regressor $h$ suffers from gradient vanishing. The main reason for this situation is that the regressor $h$ has not learned the distribution of the label, but the rejector $r$ has converged, which means that the regressor is not ready. Fortunately, we can avoid such a situation by training the rejector after the regressor is ready, and we name such a method Slow-Start. Specifically, Slow-Start prioritizes training the regressor $h$ without training the rejector $r$, and then co-trains the regressor $h$ and rejector $r$ when the regressor $h$ is capable of making predictions.

### 5.2    Datasets and Backbone Models

We conduct experiments on seven datasets, including one computer vision dataset (AgeDB [35]), one healthcare dataset (BreastPathQ [33]), and five datasets from the UCI Machine Learning Repository [13] (Abalone, Airfoil, Auto-mpg, Housing and Concrete). For each dataset, we randomly split the original dataset into training, validation, and test sets by the proportions of 60%, 20%, and 20%, respectively. It is worth noting that our approach has no restrictions on the regressor $h$ and rejector $r$, so $h$ and $r$ can be two separate parts or share parameters.

AgeDB is a regression dataset on age prediction [21] collected by [35]. It contains 16.4K face images with a minimum age of 0 and a maximum age of 101. Age prediction is not an easy task, especially when only a single photo is available. Lighting, clothing, makeup, and facial expressions all tend to affect the intuitive age, and even friends can hardly say they can identify the age in a photo. Rejecting predictions for photos with complex environments can avoid large errors. We employ ResNet-50 [25] as our backbone network for AgeDB, and the regressor $h$ and rejector $r$ share parameters. We use the Adam optimizer to train our method for 100 epochs where the Slow-Start is set to 40 epochs, the initial learning rate of $10^{-3}$ and fix the batch size to 256.

**Table 1:** Test performance (mean and std) of our surrogate loss equipped MAE on BreastPathQ. We repeat the sampling-and-training process 5 times. The metrics Rej, AR, RA are scaled to 0-100 and Sup, RcRLoss, AL and RL are all magnified by a factor of 1000.

| Cost | Sup | RcRLoss | AL | RL | Rej | AR | RA |
|------|-----|---------|-----|-----|-----|-----|-----|
| 5 | | 4.37 (0.17) | 2.70 (1.07) | 31.51 (2.29) | 72.53 (4.44) | 52.61 (5.10) | 6.53 (2.66) |
| 10 | | 8.22 (0.70) | 5.50 (1.98) | 37.14 (4.43) | 60.08 (4.34) | 43.14 (4.49) | 11.01 (4.22) |
| 15 | 16.77 (1.22) | 11.11 (0.55) | 6.84 (1.43) | 40.39 (1.67) | 53.49 (3.39) | 38.39 (2.86) | 15.46 (3.97) |
| 20 | | 13.84 (0.62) | 9.53 (1.69) | 43.41 (5.34) | 40.65 (7.28) | 29.98 (6.58) | 29.02 (9.81) |
| 25 | | 16.01 (1.32) | 12.91 (2.48) | 46.62 (9.43) | 24.47 (4.26) | 17.46 (4.96) | 48.97 (8.00) |

**Table 2:** Test performance (mean and std) of our surrogate loss equipped MAE on AgeDB. We repeat the sampling-and-training process 5 times. The metrics Rej, AR and RA are scaled to 0-100.

| Cost | Sup | RcRLoss | AL | RL | Rej | AR | RA |
|------|-----|---------|-----|-----|-----|-----|-----|
| 60 | | 59.80 (0.31) | 54.25 (4.41) | 156.81 (23.21) | 95.40 (2.88) | 93.13 (4.30) | 2.51 (1.56) |
| 70 | | 69.00 (0.39) | 61.56 (4.10) | 151.04 (12.05) | 86.22 (2.94) | 81.41 (3.07) | 8.12 (2.49) |
| 80 | | 77.10 (1.72) | 67.32 (2.21) | 150.52 (12.36) | 76.00 (15.71) | 70.63 (16.36) | 16.11 (13.20) |
| 90 | 100.34 (3.73) | 85.36 (2.23) | 73.07 (3.21) | 162.44 (12.45) | 73.38 (11.50) | 67.33 (12.07) | 17.20 (9.08) |
| 100 | | 92.94 (3.02) | 82.89 (7.47) | 170.04 (20.53) | 58.35 (12.51) | 52.15 (11.59) | 30.56 (12.48) |
| 110 | | 95.08 (5.62) | 79.62 (5.44) | 166.07 (13.75) | 52.15 (14.96) | 46.13 (14.76) | 34.38 (13.40) |
| 120 | | 96.80 (7.45) | 82.44 (2.40) | 173.14 (12.58) | 37.11 (22.64) | 32.54 (21.42) | 51.31 (23.96) |

BreastPathQ [33] is a healthcare dataset collected at the Sunnybrook Health Sciences Centre, Toronto. The dataset contains 2579 patch images, each patch has been assigned a tumor cellularity score score of 0 to 1 by 1 expert pathologist. Currently, this task is performed manually and relies upon expert interpretation of complex tissue structures. Moreover, cancer cellularity scoring is extremely risky and the use of automated methods could lead to irreversible disasters. Regression with rejection can improve this problem very well by predicting only the accepted samples and leaving the rejected samples back to the experts for evaluation. We use the same network as AgeDB and train 300 epochs using Adam optimizer where the Slow-Start is set to 50 epochs, the initial learning rate of $10^{-3}$ and fix the batch size to 128.

We conducted experiments on five UCI benchmark datasets including Abalone, Airfoil, Auto-mpg, Housing and Concrete. All of these datasets can be downloaded from the UCI Machine Learning [13]. Since our proposed method do not depend on a specific model, and we train two types of base models including the linear model and the multilayer perceptron (MLP) to support the flexibility of our method on choosing a model, where the MLP model is a five-layer ($d$-20-30-10-1) neural network with a ReLU activation function. For the rejector $r$ and regressor $h$, we consider them as two separate parts with the same structure. For both the linear model and the MLP model, we use the Adam optimization method with the batch size set to 1024 and the number of training epochs set to 1000 where the Slow-Start is set to 200 epochs. The learning rate for all UCI benchmark datasets is selected from $\{10^{-1}, 10^{-2}, 10^{-3}\}$.

### 5.3 Evaluation Metrics

For evaluation metrics, we use the RcR loss (RcRLoss) in Eq. (6) and the rejection rate (Rej). In order to further investigate how the model work, we propose additional metrics. Accepted losses

**Table 3:** Test performance (mean and std) of our surrogate loss equipped MAE on five UCI datasets trained with the MLP model. We repeat the sampling-and-training process 10 times. The metrics Rej, AR, and RA are scaled to 0-100.

| Datasets | Cost | Sup | RcRLoss | AL | RL | Rej | AR | RA |
|---|---|---|---|---|---|---|---|---|
| Abalone | 3 |  | 2.41 (0.12) | 1.99 (0.21) | 8.13 (1.08) | 42.04 (3.18) | 32.82 (3.44) | 33.33 (3.22) |
|  | 4 | 4.44 (0.46) | 2.88 (0.13) | 2.30 (0.21) | 11.37 (1.70) | 33.70 (2.47) | 25.56 (2.81) | 39.27 (3.71) |
|  | 5 |  | 3.22 (0.23) | 2.66 (0.35) | 10.30 (1.25) | 23.43 (2.94) | 16.83 (2.41) | 48.98 (5.90) |
|  | 6 |  | 3.53 (0.25) | 2.93 (0.35) | 12.13 (1.69) | 19.32 (3.47) | 13.81 (3.33) | 53.20 (5.67) |
| Airfoil | 9 |  | 7.20 (0.35) | 4.23 (0.86) | 37.80 (2.95) | 62.23 (3.73) | 41.49 (5.73) | 11.60 (3.29) |
|  | 12 |  | 8.11 (0.36) | 5.39 (0.86) | 51.51 (10.51) | 40.33 (7.95) | 23.37 (7.20) | 25.88 (9.04) |
|  | 16 | 12.96 (2.60) | 9.15 (0.43) | 6.84 (0.70) | 72.80 (20.79) | 24.92 (5.67) | 11.92 (6.93) | 38.17 (5.02) |
|  | 20 |  | 11.32 (0.75) | 8.83 (1.47) | 58.28 (8.87) | 21.53 (7.71) | 13.70 (5.34) | 48.66 (18.08) |
|  | 25 |  | 11.47 (1.54) | 9.24 (1.35) | 74.38 (16.07) | 14.19 (5.11) | 8.08 (3.60) | 52.11 (12.32) |
|  | 30 |  | 11.68 (3.07) | 11.17 (3.20) | 96.55 (16.60) | 2.52 (3.81) | 1.38 (1.78) | 86.35 (20.06) |
| Auto-mpg | 4 |  | 3.64 (0.29) | 2.99 (0.83) | 13.98 (4.16) | 56.92 (13.00) | 46.80 (15.49) | 28.74 (10.51) |
|  | 6 |  | 4.83 (0.93) | 3.83 (1.70) | 18.04 (5.95) | 37.31 (14.10) | 29.01 (12.74) | 42.42 (19.54) |
|  | 8 | 8.34 (2.16) | 6.75 (1.93) | 6.14 (2.41) | 25.59 (12.48) | 22.95 (19.88) | 19.26 (18.27) | 64.99 (23.95) |
|  | 10 |  | 7.14 (1.64) | 6.11 (2.24) | 23.29 (9.54) | 24.07 (6.58) | 17.15 (5.12) | 48.47 (15.98) |
|  | 13 |  | 8.13 (2.41) | 7.42 (2.83) | 35.49 (23.74) | 12.56 (6.83) | 10.38 (6.14) | 71.52 (14.52) |
| Housing | 9 |  | 8.80 (0.34) | 6.25 (3.22) | 40.28 (17.30) | 84.46 (11.67) | 77.60 (15.88) | 9.72 (5.91) |
|  | 12 | 12.57 (3.43) | 9.52 (0.75) | 7.40 (1.48) | 58.94 (25.98) | 44.65 (8.69) | 33.30 (8.99) | 31.25 (8.64) |
|  | 16 |  | 10.12 (1.84) | 8.35 (1.58) | 88.14 (44.53) | 22.38 (8.90) | 14.21 (6.81) | 51.84 (14.41) |
|  | 20 |  | 10.50 (3.32) | 9.59 (3.50) | 184.24 (109.35) | 8.51 (6.82) | 5.81 (5.32) | 73.40 (13.11) |
| Concrete | 20 |  | 18.03 (1.32) | 13.17 (4.91) | 82.17 (14.58) | 69.42 (6.92) | 54.06 (9.37) | 12.34 (4.47) |
|  | 30 |  | 24.20 (1.85) | 19.29 (3.85) | 112.13 (30.32) | 44.08 (8.81) | 27.43 (8.55) | 26.80 (7.90) |
|  | 40 | 34.44 (3.05) | 28.63 (2.56) | 23.12 (4.59) | 136.51 (46.59) | 31.50 (8.98) | 18.32 (7.30) | 39.49 (12.07) |
|  | 50 |  | 32.48 (2.76) | 27.90 (4.31) | 168.19 (41.73) | 19.76 (7.54) | 10.54 (4.51) | 53.82 (13.74) |
|  | 60 |  | 34.33 (3.50) | 30.33 (4.89) | 197.26 (49.03) | 12.82 (6.62) | 5.67 (3.21) | 60.95 (14.99) |

(AL) and rejected losses (RL) denote losses on accepted instances and rejected instances, and they are defined as $\frac{\sum_{i=1}^{n} \mathbb{I}[r(\boldsymbol{x}_i)>0](h(\boldsymbol{x}_i)-y_i)^2}{\sum_{i=1}^{n} \mathbb{I}[r(\boldsymbol{x}_i)>0]}$ and $\frac{\sum_{i=1}^{n} \mathbb{I}[r(\boldsymbol{x}_i)\leq 0](h(\boldsymbol{x}_i)-y_i)^2}{\sum_{i=1}^{n} \mathbb{I}[r(\boldsymbol{x}_i)\leq 0]}$. We also present the false rejection rate (AR) and false acceptance rate (RA) similar to false negative and false positive, which denote the rate of instances that should be accepted that are rejected and the ratio of instances that should be rejected that are accepted, and they are defined as $\frac{\sum_{i=1}^{n} \mathbb{I}[(h(\boldsymbol{x}_i)-y_i)^2<c(\boldsymbol{x}_i)]\mathbb{I}[r(\boldsymbol{x}_i)\leq 0]}{\sum_{i=1}^{n} \mathbb{I}[(h(\boldsymbol{x}_i)-y_i)^2<c(\boldsymbol{x}_i)]}$ and $\frac{\sum_{i=1}^{n} \mathbb{I}[(h(\boldsymbol{x}_i)-y_i)^2\geq c(\boldsymbol{x}_i)]\mathbb{I}[r(\boldsymbol{x}_i)>0]}{\sum_{i=1}^{n} \mathbb{I}[(h(\boldsymbol{x}_i)-y_i)^2\geq c(\boldsymbol{x}_i)]}$. It is worth noting that the optimal pair $(h^\star, r^\star)$ is unknown, so AR and RA are for the current regressor and rejector. We also provide the results under supervised regression method (Sup) that directly trains the model with MSE from fully training set.

## 5.4 Formulation of Surrogates and Setting of Rejection Costs

In our experiments, we consider a variety of binary classification loss functions, such as mean absolute error (MAE), mean square loss, logistic loss, sigmoid and hinge loss. The rejection cost

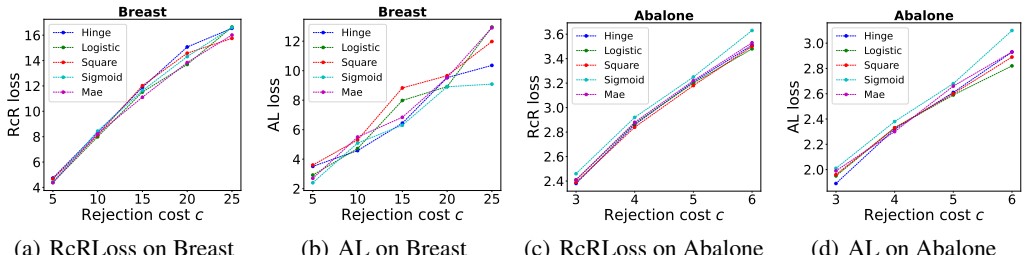

| (a) RcRLoss on Breast | (b) AL on Breast | (c) RcRLoss on Abalone | (d) AL on Abalone |

**Figure 1:** Figures (a) and (b) report the RcRLoss and AL with different rejection cost when different binary classification losses are equipped to the Breast dataset, respectively. Figures (c) and (d) report the RcRLoss and AL with different rejection costs when different binary classification losses are equipped to the Abalone dataset, respectively.

$c(\boldsymbol{x})$ is considered as a constant, which is the most commonly considered scenario in learning with rejection [6, 8, 36, 11]. For each dataset, we set various rejection costs $c$ including extreme cases and unstressed cases depending on the supervised loss. The *complete* experiments are provided in Appendix D.

### 5.5 Experimental Performance

Table 1, Table 2 and Table 3 show the experimental results when our surrogate loss function equipped with MAE on the AgeDB, BreastPathQ, and UCI datasets, respectively. From these three tables, we have the following observations: (1) Our proposed method significantly outperforms the supervised regression method in almost all cases, which validates the effectiveness of our method by rejecting difficult test instances. (2) In most cases, the average loss of our method in accepted test instances is always smaller than the average loss of the supervised regression model in all test instances. This further indicates the ability of our method to identify hard-to-predict instances and reject them. (3) As the rejection cost $c$ increases, we can clearly see the following trends in all datasets: RcRLoss decreases; Rejection rate decrease; Accepted loss increases. This is because as the prediction error we can accept increases, i.e., the rejection cost increases, the rejector will accept more instances, leading to a decrease in the rejection rate. However, the regressor capacity remains the same and more instances (containing difficult instances) also face more challenges, so RcRLoss and AL increase but remain smaller than Sup. (4) In setting the rejection cost $c$, we consider many extreme cases, e.g., the rejection cost is much smaller and much larger than the average loss in the supervised regression. In such extreme cases, our approach is still effective to identify and reject difficult test instances. In addition, we show in Figure 1 a plot of RcRLoss with the increasing rejection cost when different binary classification losses are equipped on the Breast and Abalone datasets. As the rejection cost $c$ increases, both AL loss and RcRLoss also increase.

### 5.6 Comparison with Selective Regression Methods

Since our paper provides the first attempt to investigate regression with cost-based rejection, there does not exist a baseline that can be directly compared. However, the Bayes optimal solution of regression with cost-based rejection is the same as the Bayes optimal solution of selective regression, so we can compare with selective regression methods. We have conducted additional experiments to compare with SelectiveNet [20] and Knn-Plug [46]. For SelectiveNet, this work proposes a neural network architecture and an optimization goal to control the rejection threshold of the rejector to ensure a specific rejection rate. To ensure a fair comparison, we use the same base model for all datasets. For Knn-Plug, this method proposes a semi-supervised learning process for learning a data-driven predictor with a reject option based on the plug-in principle. Specifically, this method learns a regression function $h$ and a conditional variance function $\sigma$ from the labeled dataset, while the unlabeled dataset is used to calibrate the conditional variance function $\sigma$ to ensure the desired rejection rate. For each dataset, we randomly split the original dataset into a labeled training set, an unlabeled training set, and a test set by the proportions of 60% (the same size of labeled training set as ours), 20%, and 20%, respectively. To ensure fairness, we replace the original base model Knn with MLP and named MLP-Plug.

**Table 4:** Comparison with SelectiveNet and MLP-Plug. The cost is the rejection cost in regression with cost-based rejection and Rj is the expected rejection rate in selective regression.

| Datasets | RcR_MAE | | | | MLP Plug | | | SelectiveNet | | |
|---|---|---|---|---|---|---|---|---|---|---|
| | Cost | RcRLoss | AL | Rej | Rj | AL | Rej | Rj | AL | Rej |
| Abalone | 3 | 2.41 (0.12) | **1.99** (**0.21**) | 42.04 (3.18) | 42.04 | 2.13 (0.22) | 40.93 (2.70) | 42.00 | 2.10 (0.19) | 42.80 (1.84) |
| | 4 | 2.88 (0.13) | **2.30** (**0.21**) | 33.70 (2.47) | 33.70 | 2.36 (0.20) | 32.30 (2.89) | 33.00 | 2.40 (0.22) | 34.26 (1.52) |
| | 5 | 3.22 (0.23) | **2.66** (**0.35**) | 23.43 (2.94) | 23.43 | 2.72 (0.23) | 22.00 (3.69) | 23.00 | 2.84 (0.42) | 25.10 (1.71) |
| | 6 | 3.53 (0.25) | 2.93 (0.35) | 19.32 (3.47) | 19.32 | 2.94 (0.31) | 17.64 (2.71) | 19.00 | **2.92** (**0.39**) | 21.20 (2.00) |
| Auto-mpg | 4 | 3.64 (0.29) | **2.99** (**0.83**) | 56.92 (13.00) | 56.92 | 4.83 (2.28) | 56.08 (6.53) | 56.00 | 3.58 (1.91) | 57.44 (5.06) |
| | 6 | 4.83 (0.93) | **3.83** (**1.70**) | 37.31 (14.10) | 37.31 | 5.64 (2.14) | 36.46 (9.47) | 37.00 | 6.29 (2.48) | 35.13 (6.34) |
| | 8 | 6.75 (1.93) | **6.14** (**2.41**) | 22.95 (19.88) | 22.95 | 6.30 (2.03) | 23.67 (9.82) | 23.00 | 6.86 (2.08) | 23.59 (5.91) |
| | 13 | 8.13 (2.41) | **7.42** (**2.83**) | 12.56 (6.83) | 12.56 | 6.68 (1.93) | 10.63 (3.54) | 13.00 | 7.79 (2.29) | 11.15 (3.49) |
| Housing | 9 | 8.80 (0.34) | **6.25** (**3.22**) | 84.46 (11.67) | 84.46 | 8.56 (4.70) | 84.90 (8.22) | 84.00 | 7.70 (2.22) | 86.88 (4.56) |
| | 12 | 9.52 (0.75) | **7.40** (**1.48**) | 44.65 (8.69) | 44.65 | 9.76 (2.82) | 44.80 (8.62) | 45.00 | 8.20 (2.11) | 46.83 (5.86) |
| | 16 | 10.12 (1.84) | **8.35** (**1.58**) | 22.38 (8.90) | 22.38 | 10.52 (3.70) | 22.55 (6.50) | 22.00 | 8.73 (1.64) | 26.44 (4.52) |
| | 20 | 10.50 (3.32) | 9.59 (3.50) | 8.51 (6.82) | 8.51 | 11.16 (3.72) | 10.20 (4.14) | 9.00 | **8.79** (**1.67**) | 11.39 (2.44) |
| Concrete | 20 | 18.03 (1.32) | **13.17** (**4.91**) | 69.42 (6.92) | 69.00 | 18.96 (5.71) | 71.17 (4.93) | 69.00 | 14.44 (4.42) | 71.60 (3.16) |
| | 30 | 24.20 (1.85) | **19.29** (**3.85**) | 44.08 (8.81) | 44.00 | 25.13 (4.57) | 48.20 (8.23) | 44.00 | 21.48 (3.77) | 49.27 (2.54) |
| | 40 | 28.63 (2.56) | **23.12** (**4.59**) | 31.50 (8.98) | 32.00 | 26.69 (4.83) | 33.74 (7.59) | 31.00 | 25.02 (1.67) | 36.99 (2.46) |
| | 50 | 32.48 (2.76) | **27.90** (**4.31**) | 19.76 (7.54) | 20.00 | 29.26 (4.56) | 21.36 (5.37) | 20.00 | 28.66 (3.23) | 28.54 (2.91) |
| | 60 | 34.33 (3.50) | **30.33** (**4.89**) | 12.82 (6.62) | 13.00 | 31.23 (4.65) | 14.90 (4.81) | 12.00 | 30.51 (3.90) | 20.00 (3.59) |

Table 4 shows the results of comparison experiments. Specifically, to be able to establish a connection between our studied RcR and selective regression, we set the expected rejection rate (Rj) of the selective regression method based on the results of RcR_MAE (our surrogate loss equipped MAE). It is important to note that there is no way to perfectly match the rejection rate, as the set Rj is "expected". In addition, we show plots of the variation in AL loss for all methods with different rejection rates in Appendix D.7. As can be seen Table 4, our proposed method outperforms (smaller AL loss with the same rejection rate) almost all compared methods, which validates the effectiveness of our method.

# 6 Conclusion

In this paper, we investigated a novel regression problem called regression with cost-based rejection, which aims to learn a model that can reject predictions to avoid critical mispredictions at a certain rejection cost. In order to solve this problem, we first formulated the expected risk for regression with cost-based rejection and derived the Bayes optimal solution for the expected risk, which shows that we should reject instances where the variance is greater than the rejection cost. Since the variance is difficult to obtain, we proposed a surrogate loss function that considers rejection as binary classification. Further, we provided conditions for the consistency of our method, implying that the optimal solution can be recovered by our method. Finally, we derived the regret transfer bound and the estimation error bound for our method and conducted extensive experiments on various datasets to demonstrate the effectiveness of our proposed method. We expect that our first study of a simple yet theoretically grounded method for regression with cost-based rejection can inspire more interesting studies on this new problem.

## Acknowledgements

Lei Feng is supported by Chongqing Overseas Chinese Entrepreneurship and Innovation Support Program, CAAI-Huawei MindSpore Open Fund, and Openl Community (https://openi.pcl.ac.cn). Bo An is supported by the National Research Foundation, Singapore under its Industry Alignment Fund – Pre-positioning (IAF-PP) Funding Initiative. Any opinions, findings and conclusions or recommendations expressed in this material are those of the author(s) and do not reflect the views of National Research Foundation, Singapore.

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

# A   Proof of Theorem 2

For an instance $\boldsymbol{x}$, we have the following expected risk for $\boldsymbol{x}$:

$$R_{\text{RcR}|\boldsymbol{x}}(h, r) = \mathbb{E}_{p(y|\boldsymbol{x})}[\mathcal{L}(h, r, c, \boldsymbol{x}, y)]$$
$$= \int_{\mathcal{Y}} p(y|\boldsymbol{x})\mathcal{L}(h, r, c, \boldsymbol{x}, y)\mathrm{d}y$$

If we refuse to make a prediction for $\boldsymbol{x}$, i.e., $r(\boldsymbol{x}) < 0$, the above expected risk transforms into the following equation:

$$R_{\text{RcR}|\boldsymbol{x}, r(\boldsymbol{x})<0}(h, r) = \int_{\mathcal{Y}} p(y|\boldsymbol{x})\mathcal{L}(h, r, c, \boldsymbol{x}, y)\mathrm{d}y$$
$$= \int_{\mathcal{Y}} p(y|\boldsymbol{x})c(\boldsymbol{x})\mathrm{d}y$$
$$= c(\boldsymbol{x}).$$

When we make a prediction for $\boldsymbol{x}$, i.e., $r(\boldsymbol{x}) > 0$, the above expected risk transforms into the following equation:

$$R_{\text{RcR}|\boldsymbol{x}, r(\boldsymbol{x})>0}(h, r) = \int_{\mathcal{Y}} p(y|\boldsymbol{x})\mathcal{L}(h, r, c, \boldsymbol{x}, y)\mathrm{d}y$$
$$= \int_{\mathcal{Y}} p(y|\boldsymbol{x})(h(\boldsymbol{x}) - y)^2\mathrm{d}y$$
$$= \int_{\mathcal{Y}} p(y|\boldsymbol{x})(h(\boldsymbol{x})^2 - 2yh(\boldsymbol{x}) + y^2)\mathrm{d}y$$
$$= \int_{\mathcal{Y}} p(y|\boldsymbol{x})h(\boldsymbol{x})^2\mathrm{d}y - \int_{\mathcal{Y}} p(y|\boldsymbol{x})2yh(\boldsymbol{x})\mathrm{d}y + \int_{\mathcal{Y}} p(y|\boldsymbol{x})y^2\mathrm{d}y$$
$$= h^2(\boldsymbol{x}) - 2h(\boldsymbol{x})\mathbb{E}_{p(y|\boldsymbol{x})}[y] + \mathbb{E}_{p(y|\boldsymbol{x})}[y^2]$$
$$= h^2(\boldsymbol{x}) - 2h(\boldsymbol{x})\mathbb{E}_{p(y|\boldsymbol{x})}[y] + \mathbb{E}^2_{p(y|\boldsymbol{x})}[y] + \mathbb{D}_{p(y|\boldsymbol{x})}[y]$$
$$= (h(\boldsymbol{x}) - \mathbb{E}_{p(y|\boldsymbol{x})}[y])^2 + \mathbb{D}_{p(y|\boldsymbol{x})}[y]$$

When $h(\boldsymbol{x}) = \mathbb{E}_{p(y|\boldsymbol{x})}[y]$ makes $R_{\text{RcR}|\boldsymbol{x}, r(\boldsymbol{x})>0} = \mathbb{D}_{p(y|\boldsymbol{x})}[y]$ minimum. It is easy to know that $R_{\text{RcR}|\boldsymbol{x}, r(\boldsymbol{x})>0} < R_{\text{RcR}|\boldsymbol{x}, r(\boldsymbol{x})<0}$ when $c(\boldsymbol{x}) - \mathbb{D}_{p(y|\boldsymbol{x})}[y] > 0$ and $R_{\text{RcR}|\boldsymbol{x}, r(\boldsymbol{x})>0} > R_{\text{RcR}|\boldsymbol{x}, r(\boldsymbol{x})<0}$ when $c(\boldsymbol{x}) - \mathbb{D}_{p(y|\boldsymbol{x})}[y] < 0$ which means that $R_{\text{RcR}|\boldsymbol{x}}$ is minimum when the following equation holds.

$$r^\star(\boldsymbol{x}) = \mathbb{I}(c(\boldsymbol{x}) - \mathbb{D}_{p(y|\boldsymbol{x})}[y]).$$

The proof is completed. $\qquad\qquad\qquad\qquad\qquad\qquad\qquad\qquad\qquad\qquad\quad$ $\square$

# B  Proofs of Consistent and Calibration

## B.1  Proof of Theorem 5

First, we prove that the optimal regressor $h^\star$ is also an optimal regressor for $R_{\mathrm{RcR}}^\psi$ as follows.

$$R_{\mathrm{RcR}}^\psi(h^\star, r)$$
$$= \mathbb{E}_{p(\boldsymbol{x},y)}[\psi(h^\star, r, c, \boldsymbol{x}, y)]$$
$$= \mathbb{E}_{p(\boldsymbol{x},y)}[(h^\star(\boldsymbol{x}) - y)^2\ell(r(\boldsymbol{x}), -1) + c(\boldsymbol{x})\ell(r(\boldsymbol{x}), +1)]$$
$$= \mathbb{E}_{p(\boldsymbol{x},y)}[(h^\star(\boldsymbol{x})^2 - 2yh^\star(\boldsymbol{x}) + y^2)\ell(r(\boldsymbol{x}), -1) + c(\boldsymbol{x})\ell(r(\boldsymbol{x}), +1)]$$
$$= \int_{\mathcal{X}} \int_{\mathcal{Y}} p(\boldsymbol{x}, y)[(h^\star(\boldsymbol{x})^2 - 2yh^\star(\boldsymbol{x}) + y^2)\ell(r(\boldsymbol{x}), -1) + c(\boldsymbol{x})\ell(r(\boldsymbol{x}), +1)]\mathrm{d}y\mathrm{d}\boldsymbol{x}$$
$$= \int_{\mathcal{X}} \int_{\mathcal{Y}} p(y|\boldsymbol{x})p(\boldsymbol{x})[(h^\star(\boldsymbol{x})^2 - 2yh^\star(\boldsymbol{x}) + y^2)\ell(r(\boldsymbol{x}), -1) + c(\boldsymbol{x})\ell(r(\boldsymbol{x}), +1)]\mathrm{d}y\mathrm{d}\boldsymbol{x}$$
$$= \int_{\mathcal{X}} p(\boldsymbol{x})[(h^\star(\boldsymbol{x})^2 - \int_{\mathcal{Y}} 2yh^\star(\boldsymbol{x})p(y|\boldsymbol{x})\mathrm{d}y + \int_{\mathcal{Y}} y^2 p(y|\boldsymbol{x})\mathrm{d}y)\ell(r(\boldsymbol{x}), -1) + c(\boldsymbol{x})\ell(r(\boldsymbol{x}), +1)]\mathrm{d}\boldsymbol{x}$$
$$= \int_{\mathcal{X}} p(\boldsymbol{x})[(h^\star(\boldsymbol{x})^2 - 2h^\star(\boldsymbol{x})\mathbb{E}_{p(y|\boldsymbol{x})}[y] + \mathbb{E}_{p(y|\boldsymbol{x})}[y^2])\ell(r(\boldsymbol{x}), -1) + c(\boldsymbol{x})\ell(r(\boldsymbol{x}), +1)]\mathrm{d}\boldsymbol{x}$$
$$= \int_{\mathcal{X}} p(\boldsymbol{x})[(h^\star(\boldsymbol{x})^2 - 2h^\star(\boldsymbol{x})\mathbb{E}_{p(y|\boldsymbol{x})}[y] + \mathbb{E}_{p(y|\boldsymbol{x})}^2[y] + \mathbb{D}_{p(y|\boldsymbol{x})}[y])\ell(r(\boldsymbol{x}), -1) + c(\boldsymbol{x})\ell(r(\boldsymbol{x}), +1)]\mathrm{d}\boldsymbol{x}$$
$$= \int_{\mathcal{X}} p(\boldsymbol{x})[((h^\star(\boldsymbol{x})^2 - \mathbb{E}_{p(y|\boldsymbol{x})}[y])^2 + \mathbb{D}_{p(y|\boldsymbol{x})}[y])\ell(r(\boldsymbol{x}), -1) + c(\boldsymbol{x})\ell(r(\boldsymbol{x}), +1)]\mathrm{d}\boldsymbol{x}, \tag{12}$$

where $h^\star(\boldsymbol{x}) = \mathbb{E}_{p(y|\boldsymbol{x})}[y]$ makes the above risk minimal when $\forall \boldsymbol{x} \in \mathcal{X}$, $\ell(r(\boldsymbol{x}), z) \geq 0$, and thus $h^\star$ is an optimal regressor for $R_{\mathrm{RcR}}^\psi$, with any rejector $r$. On the other hand, we prove that $h^\star$ is the only optimal regressor if the condition: $\forall \boldsymbol{x} \in \mathcal{X}$, $\ell(r(\boldsymbol{x}), z) > 0$ is achieved.

First we prove that $h^*$ is not the only optimal regressor when the condition: $\forall \boldsymbol{x} \in \mathcal{X}$, $\ell(r(\boldsymbol{x}), z) \geq 0$ is achieved by contradiction. Specifically, we assume that there is at least one other regressor that makes risk minimize. Suppose given an instance $\boldsymbol{x}_0$ and a rejector $r'$ such that $\ell(r'(\boldsymbol{x}_0), -1) = 0$, hence we have at least one other regressor $h'$ such that $R_{\mathrm{RcR}}^\psi(h', r') = R_{\mathrm{RcR}}^\psi(h^\star, r')$ and $h'(\boldsymbol{x}_0) \neq h^\star(\boldsymbol{x}_0)$ due to the following equation holds:

$$\mathbb{D}_{p(y|\boldsymbol{x}_0)}[y]\ell(r'(\boldsymbol{x}_0), -1) = 0. \tag{13}$$

The above equation implies that there exist multiple regressors that minimize the risk when the condition: $\forall \boldsymbol{x} \in \mathcal{X}$, $\ell(r(\boldsymbol{x}), z) \geq 0$ is achieved. This is mainly due to the binary classification loss $\ell(r(\boldsymbol{x}), z) = 0$. However, we can easily know that $h^*$ is the only optimal regressor when the condition: $\forall \boldsymbol{x} \in \mathcal{X}$, $\ell(r(\boldsymbol{x}), z) > 0$ is achieved, since there are no ignored terms in Eq. (12). □

## B.2  Proof of Theorem 6

Let us go back to the discussion of Eq. (12):

$$R_{\mathrm{RcR}}^\psi(h^\star, r) = \int_{\mathcal{X}} [((h^\star(\boldsymbol{x})^2 - \mathbb{E}_{p(y|\boldsymbol{x})}[y])^2 + \mathbb{D}_{p(y|\boldsymbol{x})}[y])\ell(r(\boldsymbol{x}), -1) + c(\boldsymbol{x})\ell(r(\boldsymbol{x}), +1)]p(\boldsymbol{x})\mathrm{d}\boldsymbol{x}$$
$$= \int_{\mathcal{X}} \mathbb{D}_{p(y|\boldsymbol{x})}[y]\ell(r(\boldsymbol{x}), -1)p(\boldsymbol{x}) + c(\boldsymbol{x})\ell(r(\boldsymbol{x}), +1)p(\boldsymbol{x})\mathrm{d}\boldsymbol{x}.$$

Similarly to the proof of Theorem 5, the optimal regressor $h^\star$ still minimizes the risk for any rejector. However, it is easy to know that for a rejector $r_0$ when there exists an instance $\boldsymbol{x}_0$ such that $\ell(r'(\boldsymbol{x}_0), -1) = 0$, there exists at least one other regressor $h'$ such that $R_{\mathrm{RcR}}^\psi(h', r') = R_{\mathrm{RcR}}^\psi(h^\star, r')$ and $h'(\boldsymbol{x}_0) \neq h^\star(\boldsymbol{x}_0)$ due to $((h(\boldsymbol{x})^2 - \mathbb{E}_{p(y|\boldsymbol{x})}[y])^2 + \mathbb{D}_{p(y|\boldsymbol{x})}[y])\ell(r(\boldsymbol{x}), -1) = 0$ always holds. Therefore, the optimal regressor $h^\star$ is not the only optimal solution. Fortunately, we can still show that it is regressor-consistent for some instances in this case.

For a binary classification loss function $\ell$, we denote by $\mathcal{X}_\ell^1$ the space where $\forall \boldsymbol{x} \in \mathcal{X}_\ell^1$, $\ell(r(\boldsymbol{x}), -1) \neq 0$ for any rejector $r$. It is worth noting that when $\ell(r(\boldsymbol{x}), -1) = 0$, we can infer $r(\boldsymbol{x}) < 0$, which means that the instance $\boldsymbol{x}$ will be rejected. Then we have the following equation:

$$R_{\mathrm{RcR}}^\psi(h, r) = \int_{\mathcal{X}} [((h(\boldsymbol{x}) - \mathbb{E}_{p(y|\boldsymbol{x})}[y])^2 + \mathbb{D}_{p(y|\boldsymbol{x})}[y])\ell(r(\boldsymbol{x}), -1) + c(\boldsymbol{x})\ell(r(\boldsymbol{x}), +1)]p(\boldsymbol{x})\mathrm{d}\boldsymbol{x}$$

$$= \int_{\mathcal{X}_\ell^1} (h(\boldsymbol{x}) - \mathbb{E}_{p(y|\boldsymbol{x})}[y])^2 \ell(r(\boldsymbol{x}), -1)p(\boldsymbol{x})\mathrm{d}\boldsymbol{x}$$

$$+ \int_{\mathcal{X}_\ell^1} \mathbb{D}_{p(y|\boldsymbol{x})}[y]\ell(r(\boldsymbol{x}), -1)p(\boldsymbol{x}) + c(\boldsymbol{x})\ell(r(\boldsymbol{x}), +1)p(\boldsymbol{x})\mathrm{d}\boldsymbol{x}.$$

When for all $\boldsymbol{x} \in \mathcal{X}_\ell^1$, we have the above risk is minimised when $h(\boldsymbol{x}) = \mathbb{E}_{p(y|\boldsymbol{x})}[y]$ for all $\boldsymbol{x} \in \mathcal{X}_\ell^1$. In this case, we obtain a similar form as the proof of Theorem 5. Therefore, we have proved that $h_\psi^\star$ is consistent (i.e., $h_\psi^\star = h^\star$) for accepted instances when the loss $\ell$ is non-negative. The proof is completed.

### B.3 Proof of Theorem 7

By fixing the optimal regressor $h^*$, let us discuss the pointwise version of Eq. (12):

$$\mathbb{E}_{p(y|\boldsymbol{x})}[\psi(h^\star, r, c, \boldsymbol{x}, y)] = \mathbb{D}_{p(y|\boldsymbol{x})}[y]\ell(r(\boldsymbol{x}), -1) + c(\boldsymbol{x})\ell(r(\boldsymbol{x}), +1).$$

The rejector $r(\boldsymbol{x})$ that minimizes the expected value is given by

$$\min_{r(\boldsymbol{x})} \mathbb{D}_{p(y|\boldsymbol{x})}[y]\ell(r(\boldsymbol{x}), -1) + c(\boldsymbol{x})\ell(r(\boldsymbol{x}), +1)$$

If the binary classification loss $\ell(r(\boldsymbol{x}), z)$ is classification-calibrated, then it is easy to see that the optimal rejector $r_\psi^\star$ should have the same sign with $\mathbb{D}_{p(y|\boldsymbol{x})}[y] - c(\boldsymbol{x})$ by the definition of classification calibration. When $\mathbb{D}_{p(y|\boldsymbol{x})}[y] < c(\boldsymbol{x})$, the optimal rejector $r_\psi^*(\boldsymbol{x})$ must have a positive sign, and when $\mathbb{D}_{p(y|\boldsymbol{x})}[y] \geq c(\boldsymbol{x})$, the optimal rejector $r_\psi^*(\boldsymbol{x})$ must have a negative sign. The proof is completed. $\square$

## C Proofs of the Regret Transfer Bound and the Estimation Error Bound

### C.1 Proof of Theorem 8

*Proof.* Without loss of generality, we discuss the regret transfer on a point $\boldsymbol{x}$'s conditional risk and it can be generalized to the whole distribution using Jensen's inequality of concave functions. We denote by $D = \mathbb{D}_{p(y|\boldsymbol{x})}[y]$, $\tilde{D} = \mathbb{E}_{p(\boldsymbol{x}, y)}[(h(\boldsymbol{x}) - y)^2]$, $\alpha = r(\boldsymbol{x})$, and $c = c(\boldsymbol{x})$. Notice we are using margin loss $\ell$, thus we denote by $\ell(\alpha, -1) = \phi(\alpha)$ and $\ell(\alpha, -1) = \phi(\alpha)$.

Then we can denote our conditional risk as $R(\tilde{D}, \alpha) = \tilde{D}\phi(-\alpha) + c\phi(\alpha)$, where $\mathbb{E}_{p(\boldsymbol{x})}[R(\tilde{D}, \alpha)] = R_{\mathrm{RcR}}(h, r)$. According to our previous discussion, $R(\tilde{D}, \alpha)$ achieves its minimum at $(D, \alpha^*)$, where $\alpha^* = \mathrm{argmin}_{\alpha \in \mathbb{R}} D\phi(-\alpha) + c\phi(\alpha)$. We can learn that $\tilde{D} \geq D$. Then we can write our point-wise regret as:

$$R(\tilde{D}, \alpha) - R(D, \alpha^*) = \tilde{D}\phi(-\alpha) + c\phi(\alpha) - D\phi(-\alpha^*) - c\phi(\alpha^*)$$

Then we can begin our proof. There are three cases that the point-wise regret of our target loss is non-zero:

- When $D < c$ but $\alpha \leq 0$, the point-wise regret of target loss is $c - D$.

  In this case:

$$
\begin{aligned}
R(\tilde{D}, \alpha) - R(D, \alpha^*) &= \tilde{D}\phi(-\alpha) + c\phi(\alpha) - D\phi(-\alpha^*) - c\phi(\alpha^*) \\
&\geq D\phi(-\alpha) + c\phi(\alpha) - D\phi(-\alpha^*) - c\phi(\alpha^*) \\
&\geq (D + c)\left[\frac{D}{D + c}\phi(-\alpha) + \frac{c}{D + c}\phi(\alpha) - \frac{D}{D + c}\phi(-\alpha^*) - \frac{c}{D + c}\phi(\alpha^*)\right] \\
&\geq (D + c)\xi^{-1}\left(\frac{c - D}{D + c}\right)
\end{aligned}
$$

- When $D > c$ but $\alpha \geq 0$, the point-wise regret of target loss is $D - c$.

  In this case, denote by $\tilde{\alpha}^* = \operatorname{argmin} R(\tilde{D}, \alpha)$, we can learn that $R(D, \alpha^*) \leq R(D, \tilde{\alpha}^*)$:

  $$
  \begin{aligned}
  R(\tilde{D}, \alpha) - R(D, \alpha^*) &\geq R(\tilde{D}, \alpha) - R(D, \tilde{\alpha}^*) \\
  &= \tilde{D}\phi(-\alpha) + c\phi(\alpha) - D\phi(-\tilde{\alpha}^*) - c\phi(\tilde{\alpha}^*) \\
  &\geq \tilde{D}\phi(-\alpha) + c\phi(\alpha) - \tilde{D}\phi(-\tilde{\alpha}^*) - c\phi(\tilde{\alpha}^*) \\
  &\geq (\tilde{D} + c)\left[\frac{\tilde{D}}{\tilde{D}+c}\phi(-\alpha) + \frac{c}{\tilde{D}+c}\phi(\alpha) - \frac{\tilde{D}}{\tilde{D}+c}\phi(-\tilde{\alpha}^*) - \frac{c}{\tilde{D}+c}\phi(\tilde{\alpha}^*)\right] \\
  &\geq (\tilde{D} + c)\xi^{-1}\left(\frac{\tilde{D}-c}{\tilde{D}+c}\right)
  \end{aligned}
  $$

- When $D < c$ and $\alpha \geq 0$, the point-wise regret of target loss is $\tilde{D} - D$.

  In this case, we should notice that $\alpha^* > 0$. We further split this case into two cases:

  - When $\tilde{D} > c$, $\tilde{\alpha}^* < 0$:

    $$
    \begin{aligned}
    R(\tilde{D}, \alpha) - R(D, \alpha^*) &= \tilde{D}\phi(-\alpha) + c\phi(\alpha) - D\phi(-\alpha^*) - c\phi(\alpha^*) \\
    &= \tilde{D}\phi(-\alpha) + c\phi(\alpha) - \tilde{D}\phi(-\tilde{\alpha}^*) - c\phi(\tilde{\alpha}^*) + \tilde{D}\phi(-\tilde{\alpha}^*) + c\phi(\tilde{\alpha}^*) - D\phi(-\alpha^*) - c\phi(\alpha^*) \\
    &\geq \left(\tilde{D}\phi(-\alpha) + c\phi(\alpha) - \tilde{D}\phi(-\tilde{\alpha}^*) - c\phi(\tilde{\alpha}^*)\right) + (D\phi(-\tilde{\alpha}^*) + c\phi(\tilde{\alpha}^*) - D\phi(-\alpha^*) - c\phi(\alpha^*)) \\
    &\geq (\tilde{D} + c)\xi^{-1}\left(\frac{\tilde{D}-c}{\tilde{D}-c}\right) + (D + c)\xi^{-1}\left(\frac{c-D}{D+c}\right)
    \end{aligned}
    $$

  - When $\tilde{D} \leq c$, we can learn $\tilde{\alpha}^* \geq 0$ denote by $\alpha' = \operatorname{argmin}_\alpha \tilde{D}\phi(\alpha) + D\phi(-\alpha)$, and we can learn $\alpha' \geq 0$:

    $$
    \begin{aligned}
    R(\tilde{D}, \alpha) - R(D, \alpha^*) &= \tilde{D}\phi(-\alpha) + c\phi(\alpha) - D\phi(-\alpha^*) - c\phi(\alpha^*) \\
    &\geq \tilde{D}\phi(-\tilde{\alpha}^*) + c\phi(\tilde{\alpha}^*) - D\phi(-\alpha^*) - c\phi(\alpha^*) \\
    &\geq (\tilde{D} + c)H\left(\frac{c}{\tilde{D}+c}\right) + (D + c)H\left(\frac{c}{\tilde{D}+c}\right)
    \end{aligned}
    $$

Notice that for sigmoid loss and hinge loss, $\xi^{-1}(\alpha) = |\alpha|$, $H(\alpha) = 2(1 - \alpha)$. For logistic loss, $\xi^{-1} = \frac{1}{2}\alpha^2$ and $H(\alpha) = -\alpha\log(\alpha) - (1 - \alpha)\log(1 - \alpha)$. Denote by $\gamma$ the point-wise regret of target loss. Using the linear bound of sigmoid loss and hinge loss, we can learn:

$$
R(\tilde{D}, \alpha) - R(D, \alpha^*) \geq \gamma.
$$

For logistic loss, the derivation is more complicated. In the first two cases:

$$
R(\tilde{D}, \alpha) - R(D, \alpha^*) \geq \frac{\tilde{D}+c}{2}\left(\frac{\gamma}{\tilde{D}+c}\right)^2,
$$

and thus $\sqrt{2(M + c)(R(\tilde{D}, \alpha) - R(D, \alpha^*))} \geq \gamma$. In the third case, we can learn:

$$
\begin{aligned}
R(\tilde{D}, \alpha) - R(D, \alpha^*) &\geq \frac{\tilde{D}+c}{2}\left(\frac{\tilde{D}-c}{\tilde{D}+c}\right)^2 + \frac{D+c}{2}\left(\frac{c-D}{D+c}\right)^2 \\
&\geq \frac{1}{2(\tilde{D}+c)}((\tilde{D}-c)^2 + (c-D)^2) \\
&\geq \frac{1}{4(\tilde{D}+c)}(\tilde{D}-D)^2
\end{aligned}
$$

and thus $2\sqrt{(M+c)(R(\tilde{D},\alpha)-R(D,\alpha^*))} \geq \gamma$. In the last case:

$$R(\tilde{D},\alpha) - R(D,\alpha^*) \geq (\tilde{D}+c)H\left(\frac{c}{\tilde{D}+c}\right) + (D+c)H\left(\frac{c}{\tilde{D}+c}\right)$$

$$\geq -c\log\left(\frac{c}{\tilde{D}+c}\right) - \tilde{D}\log\left(\frac{\tilde{D}}{\tilde{D}+c}\right) + c\log\left(\frac{c}{D+c}\right) + D\log\left(\frac{D}{D+c}\right)$$

$$= c\log\left(1+\frac{\tilde{D}-D}{D+c}\right) + \tilde{D}\log\left(1+\frac{c}{\tilde{D}}\right) - D\log\left(1+\frac{c}{D}\right)$$

$$\geq c\log\left(1+\frac{\tilde{D}-D}{D+c}\right)$$

$$\geq c * \frac{\frac{\tilde{D}-D}{D+c}}{1+\frac{\tilde{D}-D}{D+c}}$$

$$= \frac{c(\tilde{D}-D)}{\tilde{D}+c}$$

$$\geq \frac{c(\tilde{D}-D)}{c+c}$$

$$= \frac{\tilde{D}-D}{2}.$$

Combining these cases, we can learn

$$\gamma \leq \max\{2(R(\tilde{D},\alpha)-R(D,\alpha^*)), 2\sqrt{(M+c)(R(\tilde{D},\alpha)-R(D,\alpha^*))}\}.$$

$\square$

## C.2 Proof of Theorem 9

**Definition 10.** *(Rademacher complexity)* Let $Z_1, \cdots, Z_n$ be n *i.i.d.* random variables drawn from a probability distribution $\mu$ and $\mathcal{F} = \{f : Z \to \mathbb{R}\}$ be a class of measurable functions. Then the expected Rademacher complexity of function class $\mathcal{F}$ is given as follows:

$$\mathfrak{R}_n(\mathcal{F}) = \mathbb{E}_{Z_1,\cdots,Z_n\sim\mu}\mathbb{E}_{\boldsymbol{\sigma}}\left[\sup_{f\in\mathcal{F}}\frac{1}{n}\sum_{i=1}^{n}\sigma_i f(Z_i)\right], \tag{14}$$

where $\sigma_1, \cdots, \sigma_n$ are the Rademacher variables that take the value from $\{-1,+1\}$ uniformly.

Then we can begin proving Theorem 9.

*Proof.* Suppose that the binary loss is bounded by $M_1$ and $\rho$-Lipschitz continuous, $|h|$, $c(\boldsymbol{x})$, $|y|$ is bounded by $M_2$ and the hypothesis space $\mathcal{H}$ and $\mathcal{R}$ is strong enough, then we can learn that the total loss is bounded by $C_1 = (4M_2^2 + M_2)M_1$, and is $L_1$-Lipschitz continuous *w.r.t.* $(h,r)$, where $L_1 = \sqrt{(4M_1^2\rho + M_1\rho)^2 + 16M_1^4 M_2^2}$. By applying the McDiarmid's inequality, it is routine to show that the following inequalities hold with probability at least $1 - \frac{\delta}{2}$, respectively:

$$\sup_{h,r\in\mathcal{H},\mathcal{R}}\left(R_{\mathrm{RcR}}^{\psi}(h,r) - \hat{R}_{\mathrm{RcR}}^{\psi}(h,r)\right) \leq \underset{\boldsymbol{x}_1,\cdots,\boldsymbol{x}_n}{\mathbb{E}}\left[\sup_{h,r\in\mathcal{H},\mathcal{R}}\left(R_{\mathrm{RcR}}^{\psi}(h,r) - \hat{R}_{\mathrm{RcR}}^{\psi}(h,r)\right)\right] + C_1\sqrt{\frac{\log\frac{2}{\delta}}{2n}},$$

$$\sup_{h,r\in\mathcal{H},\mathcal{R}}\left(\hat{R}_{\mathrm{RcR}}^{\psi}(h,r) - R_{\mathrm{RcR}}^{\psi}(h,r)\right) \leq \underset{\boldsymbol{x}_1,\cdots,\boldsymbol{x}_n}{\mathbb{E}}\left[\sup_{h,r\in\mathcal{H},\mathcal{R}}\left(\hat{R}_{\mathrm{RcR}}^{\psi}(h,r) - R_{\mathrm{RcR}}^{\psi}(h,r)\right)\right] + C_1\sqrt{\frac{\log\frac{2}{\delta}}{2n}}.$$

By applying Talagrand's contraction lemma [34], we can learn that:

$$\underset{\boldsymbol{x}_1,\cdots,\boldsymbol{x}_n}{\mathbb{E}}\left[\sup_{h,r\in\mathcal{H},\mathcal{R}}\left(\hat{R}_{\mathrm{RcR}}^{\psi}(h,r) - R_{\mathrm{RcR}}^{\psi}(h,r)\right)\right] \leq \sqrt{2}L_1(\mathfrak{R}_n(\mathcal{H}) + \mathfrak{R}_n(\mathcal{R}))$$

and this conclusion also holds for another direction. Plugging this conclusion into the former inequalities and using the union bound, we can learn this inequality holds with probability at least $1 - \delta$:

$$\sup_{h,r \in \mathcal{H}, \mathcal{R}} \left| \hat{R}^{\psi}_{\text{RcR}}(h,r) - R^{\psi}_{\text{RcR}}(h,r) \right| \leq \sqrt{2} L_1 (\mathfrak{R}_n(\mathcal{H}) + \mathfrak{R}_n(\mathcal{R})) + C_1 \sqrt{\frac{\log \frac{2}{\delta}}{2n}}$$

According to the definition of empirical risk minimization and identifiable condition, we can get the following conclusion:

$$R^{\psi}_{\text{RcR}}(\hat{h},\hat{r}) - \min_{h,r \in \mathcal{H}, \mathcal{R}} R^{\psi}_{\text{RcR}}(h,r) = R^{\psi}_{\text{RcR}}(\hat{h},\hat{r}) - R^{\psi*}_{\text{RcR}}(h^*,r^*)$$

$$= \left( R^{\psi}_{\text{RcR}}(\hat{h},\hat{r}) - \hat{R}^{\psi}_{\text{RcR}}(\hat{h},\hat{r}) \right) + \left( \hat{R}^{\psi}_{\text{RcR}}(\hat{h},\hat{r}) - \hat{R}^{\psi}_{\text{RcR}}(h^*,r^*) \right) + \left( \hat{R}^{\psi}_{\text{RcR}}(h^*,r^*) - R^{\psi*}_{\text{RcR}}(h^*,r^*) \right)$$

$$\leq \left( R^{\psi}_{\text{RcR}}(\hat{h},\hat{r}) - \hat{R}^{\psi}_{\text{RcR}}(\hat{h},\hat{r}) \right) + \left( \hat{R}^{\psi}_{\text{RcR}}(h^*,r^*) - R^{\psi*}_{\text{RcR}}(h^*,r^*) \right)$$

$$\leq 2 \sup_{h,r \in \mathcal{H}, \mathcal{R}} \left| \hat{R}^{\psi}_{\text{RcR}}(h,r) - R^{\psi}_{\text{RcR}}(h,r) \right|.$$

Combining Theorem 7, we can conclude the proof. $\square$

**Table 5:** Test performance (mean and std) of our surrogate loss equipped hinge loss on BreastPathQ. We repeat the sampling-and-training process 5 times. The metrics Rej, AR, RA are scaled to 0-100 and Sup, RcRLoss, AL, and RL are all magnified by a factor of 1000.

| Cost | Sup | Rej | AL | RL | Rej | AR | RA |
|------|-----|-----|-----|-----|-----|-----|-----|
| 5 | | 4.74 (0.38) | 3.51 (1.96) | 53.05 (20.33) | 80.86 (4.17) | 60.37 (6.70) | 4.19 (2.36) |
| 10 | | 8.32 (0.21) | 4.58 (1.74) | 58.90 (13.15) | 68.99 (4.71) | 46.63 (6.45) | 5.91 (2.54) |
| 15 | 16.77 | 11.89 (0.31) | 6.44 (1.82) | 49.42 (8.12) | 62.45 (4.76) | 46.86 (5.12) | 10.69 (3.84) |
| 20 | (1.22) | 15.07 (0.33) | 9.53 (1.15) | 49.58 (8.10) | 52.33 (3.94) | 38.04 (3.47) | 17.88 (5.36) |
| 25 | | 16.54 (0.78) | 10.36 (2.39) | 58.39 (19.85) | 41.23 (8.36) | 29.71 (7.36) | 25.34 (12.36) |

**Table 6:** Test performance (mean and std) of our surrogate loss equipped hinge loss on AgeDB. We repeat the sampling-and-training process 5 times. The metrics Rej, AR, RA are scaled to 0-100.

| Cost | Sup | Rej | AL | RL | Rej | AR | RA |
|------|-----|-----|-----|-----|-----|-----|-----|
| 60 | | 59.99 (0.10) | 44.80 (13.97) | 177.36 (40.19) | 97.30 (2.16) | 95.84 (3.19) | 1.43 (1.17) |
| 70 | | 70.24 (0.50) | 71.81 (4.61) | 185.68 (26.75) | 92.41 (1.20) | 88.90 (1.75) | 4.32 (0.74) |
| 80 | | 79.67 (1.40) | 76.43 (12.86) | 185.14 (18.51) | 87.23 (2.08) | 82.63 (2.63) | 7.83 (1.88) |
| 90 | 100.34 (3.73) | 88.71 (1.08) | 76.78 (8.93) | 166.84 (5.90) | 83.43 (11.01) | 79.46 (12.07) | 11.19 (9.13) |
| 100 | | 96.95 (0.67) | 77.02 (7.46) | 182.70 (13.20) | 84.78 (6.77) | 80.39 (7.29) | 9.14 (5.49) |
| 110 | | 104.29 (0.31) | 85.84 (6.98) | 192.05 (26.75) | 73.52 (10.39) | 67.73 (10.33) | 17.41 (9.56) |
| 120 | | 111.54 (2.23) | 92.59 (4.79) | 186.50 (13.73) | 67.31 (11.60) | 61.11 (11.56) | 21.74 (10.43) |

**Table 7:** Test performance (mean and std) of our surrogate loss equipped logistic loss on five UCI datasets trained with the Linear model. We repeat the sampling-and-training process 10 times. The metrics Rej, AR, and RA are scaled to 0-100.

| Datasets | Cost | Supervised | Rej | AL | RL | Rej | AR | RA |
|---|---|---|---|---|---|---|---|---|
| Abalone | 3 | 4.92 (0.51) | 2.52 (0.08) | 1.94 (0.21) | 7.84 (1.06) | 54.77 (2.66) | 44.76 (3.45) | 24.00 (1.93) |
| | 4 | | 2.99 (0.11) | 2.39 (0.19) | 9.83 (1.44) | 36.93 (2.78) | 27.94 (3.02) | 36.55 (2.83) |
| | 5 | | 3.38 (0.18) | 2.80 (0.25) | 11.78 (1.86) | 25.90 (2.27) | 18.43 (2.08) | 46.24 (3.20) |
| | 6 | | 3.69 (0.26) | 3.19 (0.31) | 13.81 (2.00) | 17.80 (1.98) | 11.89 (1.45) | 55.08 (4.19) |
| Airfoil | 9 | 23.32 (1.54) | 8.83 (0.35) | 7.55 (2.64) | 26.44 (1.99) | 85.58 (6.10) | 78.07 (8.53) | 7.24 (3.99) |
| | 12 | | 11.31 (0.49) | 8.76 (2.18) | 27.59 (2.06) | 79.93 (3.65) | 71.90 (5.57) | 9.74 (1.97) |
| | 16 | | 14.46 (0.51) | 10.90 (1.46) | 30.60 (2.13) | 69.47 (6.49) | 60.88 (7.17) | 17.14 (5.66) |
| | 20 | | 17.10 (0.83) | 13.01 (1.79) | 33.51 (4.17) | 58.54 (5.07) | 50.38 (5.67) | 26.19 (6.54) |
| | 25 | | 19.62 (1.26) | 15.95 (2.52) | 35.26 (3.40) | 39.30 (5.76) | 34.28 (5.41) | 47.29 (8.82) |
| | 30 | | 20.94 (1.84) | 17.60 (3.18) | 42.36 (7.32) | 25.38 (8.07) | 21.35 (6.97) | 61.14 (12.70) |
| Auto-mpg | 4 | 11.66 (2.26) | 3.92 (0.26) | 2.78 (1.68) | 15.05 (4.12) | 79.87 (11.10) | 73.18 (14.13) | 15.28 (7.25) |
| | 6 | | 5.73 (0.70) | 5.25 (1.63) | 17.98 (6.14) | 58.97 (8.50) | 52.23 (9.69) | 32.06 (10.05) |
| | 8 | | 6.73 (0.52) | 5.72 (0.92) | 21.58 (7.67) | 42.56 (7.84) | 36.08 (8.66) | 43.16 (11.24) |
| | 10 | | 7.37 (0.95) | 5.94 (1.61) | 26.05 (11.45) | 31.28 (12.77) | 25.72 (10.98) | 53.96 (21.76) |
| | 13 | | 8.75 (1.64) | 7.72 (1.94) | 28.93 (12.64) | 19.62 (4.69) | 17.31 (4.60) | 69.71 (13.77) |
| Housing | 9 | 24.08 (5.34) | 8.65 (0.75) | 6.95 (3.16) | 33.87 (12.62) | 67.92 (11.51) | 58.56 (14.04) | 19.28 (7.94) |
| | 12 | | 10.27 (1.08) | 8.19 (2.90) | 40.93 (16.74) | 58.32 (10.20) | 48.25 (13.11) | 23.32 (5.75) |
| | 16 | | 12.34 (1.14) | 9.20 (2.03) | 50.31 (19.14) | 45.35 (6.04) | 36.24 (6.00) | 31.42 (6.77) |
| | 20 | | 14.19 (1.67) | 10.48 (2.72) | 55.08 (23.26) | 38.42 (6.08) | 32.22 (6.62) | 42.45 (12.64) |
| Concrete | 20 | 111.12 (8.01) | 19.80 (0.29) | 10.00 (6.44) | 204.20 (63.34) | 97.57 (2.04) | 95.25 (4.29) | 1.55 (0.86) |
| | 30 | | 29.51 (0.92) | 24.08 (12.35) | 227.13 (99.54) | 91.17 (4.57) | 87.60 (6.54) | 6.02 (3.11) |
| | 40 | | 38.09 (1.38) | 28.95 (7.18) | 282.98 (96.63) | 80.34 (6.60) | 73.83 (7.46) | 12.05 (6.39) |
| | 50 | | 46.94 (1.82) | 34.22 (9.57) | 242.13 (98.34) | 75.34 (10.15) | 68.94 (11.79) | 15.98 (7.51) |
| | 60 | | 51.96 (2.08) | 41.36 (5.76) | 370.24 (113.24) | 56.26 (2.61) | 45.96 (2.24) | 25.26 (4.63) |

# D    Additional Information of Experiments

## D.1    Evaluation Metrics

We describe in detail all the evaluation metrics we used in our experiments.

**RcR loss.** The RcR loss (RcRLoss) is the main evaluation metric for RcR. For a given example $(\boldsymbol{x}, y)$ and the rejection cost $c(\boldsymbol{x})$, the RcRLoss defined as if $r(\boldsymbol{x}) > 0$, $\mathcal{L}(h, r, c, \boldsymbol{x}, y) = (h(\boldsymbol{x}) - y)^2$, otherwise $\mathcal{L}(h, r, c, \boldsymbol{x}, y) = c(\boldsymbol{x})$.

**Rejection rate.** The rejection rate (Rej) is defined as $\frac{\sum_{i=1}^{n} \mathbb{I}[r(\boldsymbol{x}) \leq 0]}{n}$. Rej indicates the rejection rate of our model on the test dataset.

**Table 8:** Test performance (mean and std) of our surrogate loss equipped logistic loss on five UCI datasets trained with the MLP model. We repeat the sampling-and-training process 10 times. The metrics Rej, AR, and RA are scaled to 0-100.

| Datasets | Cost | Supervised | Rej | AL | RL | Rej | AR | RA |
|---|---|---|---|---|---|---|---|---|
| Abalone | 3 | | 2.41 (0.10) | 1.95 (0.18) | 8.34 (0.87) | 43.14 (2.84) | 33.37 (3.34) | 33.32 (2.81) |
| | 4 | 4.44 (0.46) | 2.87 (0.18) | 2.33 (0.29) | 9.68 (1.18) | 32.29 (3.28) | 24.28 (3.29) | 41.96 (3.29) |
| | 5 | | 3.20 (0.20) | 2.59 (0.29) | 10.86 (1.34) | 25.25 (2.15) | 17.86 (2.02) | 45.15 (3.61) |
| | 6 | | 3.48 (0.25) | 2.82 (0.35) | 11.54 (1.53) | 20.44 (1.72) | 14.74 (1.42) | 51.48 (4.63) |
| Airfoil | 9 | | 6.78 (0.47) | 5.05 (0.82) | 51.45 (4.32) | 43.68 (5.19) | 20.35 (5.21) | 23.75 (3.96) |
| | 12 | | 7.96 (0.64) | 5.79 (0.82) | 59.74 (3.53) | 35.05 (3.02) | 12.94 (2.79) | 26.90 (3.03) |
| | 16 | 12.96 (2.60) | 9.04 (0.56) | 7.46 (0.47) | 68.20 (10.78) | 18.57 (4.08) | 7.36 (3.14) | 48.00 (7.21) |
| | 20 | | 9.64 (0.74) | 7.81 (0.44) | 74.27 (10.19) | 15.05 (7.76) | 5.83 (1.91) | 47.91 (11.27) |
| | 25 | | 10.47 (1.08) | 8.50 (0.52) | 80.28 (27.67) | 12.03 (4.43) | 3.78 (1.89) | 49.00 (17.65) |
| | 30 | | 10.95 (1.11) | 8.95 (0.78) | 89.30 (31.58) | 9.50 (3.86) | 2.93 (1.10) | 50.67 (18.37) |
| Auto-mpg | 4 | | 3.85 (0.56) | 3.22 (1.51) | 11.99 (3.81) | 62.44 (10.72) | 54.18 (9.53) | 25.19 (13.29) |
| | 6 | | 5.33 (0.82) | 4.67 (1.36) | 15.08 (3.83) | 43.01 (16.01) | 35.19 (16.09) | 41.25 (15.35) |
| | 8 | 8.34 (2.16) | 6.53 (1.18) | 5.86 (1.53) | 19.34 (7.60) | 29.49 (13.45) | 23.19 (13.60) | 53.57 (14.78) |
| | 10 | | 7.06 (1.60) | 6.42 (1.91) | 21.71 (8.88) | 17.95 (3.63) | 14.15 (3.48) | 65.59 (11.17) |
| | 13 | | 7.80 (1.90) | 7.04 (2.09) | 28.55 (14.96) | 13.59 (5.35) | 11.05 (5.32) | 70.15 (14.83) |
| Housing | 9 | | 8.60 (2.49) | 8.43 (3.15) | 45.60 (27.02) | 26.57 (5.78) | 19.05 (5.63) | 66.95 (10.39) |
| | 12 | 12.57 (3.43) | 9.50 (1.56) | 8.63 (2.10) | 63.52 (26.15) | 25.44 (6.43) | 17.38 (5.84) | 52.68 (12.58) |
| | 16 | | 9.30 (1.37) | 8.03 (1.70) | 90.19 (38.08) | 15.45 (4.30) | 10.84 (3.87) | 62.99 (9.05) |
| | 20 | | 9.67 (1.40) | 8.33 (1.77) | 103.55 (54.73) | 11.18 (2.97) | 8.71 (2.70) | 70.06 (8.59) |
| Concrete | 20 | | 18.65 (1.41) | 14.32 (4.80) | 58.33 (15.88) | 68.93 (13.47) | 57.72 (17.08) | 16.19 (9.45) |
| | 30 | | 25.64 (2.50) | 23.16 (4.95) | 80.43 (14.47) | 32.85 (12.82) | 19.30 (10.81) | 48.23 (15.91) |
| | 40 | 34.44 (3.05) | 29.79 (2.33) | 25.25 (3.55) | 107.54 (22.18) | 30.24 (8.58) | 19.97 (7.91) | 44.38 (9.87) |
| | 50 | | 31.63 (4.29) | 25.79 (4.22) | 120.02 (24.12) | 24.22 (11.22) | 15.29 (10.47) | 47.42 (10.35) |
| | 60 | | 34.04 (4.48) | 33.26 (4.55) | 165.71 (62.18) | 2.77 (5.13) | 2.14 (2.93) | 92.59 (12.85) |

**Accepted loss.** The accepted loss (AL) is defined as $\frac{\sum_{i=1}^{n} \mathbb{I}[r(\boldsymbol{x}_i)>0](h(\boldsymbol{x}_i)-y_i)^2}{\sum_{i=1}^{n} \mathbb{I}[r(\boldsymbol{x}_i)>0]}$. AL denotes the average loss of our regressor on the accepted test dataset.

**Rejected loss.** The rejected loss (RL) is defined as $\frac{\sum_{i=1}^{n} \mathbb{I}[r(\boldsymbol{x}_i)\leq0](h(\boldsymbol{x}_i)-y_i)^2}{\sum_{i=1}^{n} \mathbb{I}[r(\boldsymbol{x}_i)\leq0]}$. RL denotes the average loss of our regressor on the rejected test dataset.

**False rejection rate.** The false rejection rate (AR) is defined as $\frac{\sum_{i=1}^{n} \mathbb{I}[(h(\boldsymbol{x}_i)-y_i)^2<c(\boldsymbol{x}_i)]\mathbb{I}[r(\boldsymbol{x}_i)\leq0]}{\sum_{i=1}^{n} \mathbb{I}[(h(\boldsymbol{x}_i)-y_i)^2<c(\boldsymbol{x}_i)]}$. AR denotes the rate of instances that should be accepted that are rejected.

**Table 9:** Test performance (mean and std) of our surrogate loss equipped logistic loss on BreastPathQ. We repeat the sampling-and-training process 5 times. The metrics Rej, AR, RA are scaled to 0-100 and Sup, RcRLoss, AL and RL are all magnified by a factor of 1000.

| Cost | Sup | RcRLoss | AL | RL | Rej | AR | RA |
|------|-----|---------|-----|-----|-----|-----|-----|
| 5 | | 4.41 (0.35) | 2.92 (1.51) | 35.59 (7.36) | 71.56 (3.20) | 47.48 (5.95) | 5.71 (3.02) |
| 10 | | 7.99 (0.47) | 4.72 (1.01) | 41.34 (9.74) | 61.72 (9.54) | 40.69 (4.90) | 10.30 (1.02) |
| 15 | 16.77 (1.22) | 11.52 (0.36) | 7.98 (1.36) | 42.67 (5.87) | 50.48 (5.85) | 35.50 (4.61) | 20.18 (7.72) |
| 20 | | 13.69 (0.81) | 8.92 (1.05) | 63.51 (49.02) | 43.65 (5.24) | 31.11 (5.00) | 22.61 (5.72) |
| 25 | | 16.64 (0.94) | 12.96 (2.28) | 35.27 (2.63) | 29.84 (5.93) | 23.63 (5.42) | 44.77 (10.54) |

**Table 10:** Test performance (mean and std) of our surrogate loss equipped square loss on BreastPathQ. We repeat the sampling-and-training process 5 times. The metrics Rej, AR, RA are scaled to 0-100 and Sup, RcRLoss, AL and RL are all magnified by a factor of 1000.

| Cost | Sup | RcRLoss | AL | RL | Rej | AR | RA |
|------|-----|---------|-----|-----|-----|-----|-----|
| 5 | | 4.67 (0.41) | 3.60 (1.17) | 36.70 (4.46) | 69.23 (4.94) | 44.09 (4.16) | 6.26 (3.25) |
| 10 | | 8.13 (0.38) | 5.30 (0.73) | 43.66 (6.95) | 59.69 (2.59) | 37.47 (2.40) | 10.42 (2.60) |
| 15 | 16.77 | 12.02 (1.09) | 8.83 (2.14) | 39.83 (8.27) | 51.70 (2.56) | 36.78 (0.80) | 17.65 (2.03) |
| 20 | (1.22) | 14.58 (0.57) | 9.66 (2.55) | 43.69 (7.58) | 44.72 (9.54) | 33.20 (7.89) | 24.21 (11.09) |
| 25 | | 15.75 (0.76) | 11.98 (2.59) | 45.65 (7.18) | 27.57 (8.62) | 19.94 (7.86) | 43.73 (12.04) |

**False acceptance rate.** The false acceptance rate (RA) denotes the rate of instances that should be rejected that are accepted, and is defined as $\frac{\sum_{i=1}^{n} \mathbb{I}[(h(\boldsymbol{x}_i)-y_i)^2 \geq c(\boldsymbol{x}_i)]\mathbb{I}[r(\boldsymbol{x}_i)>0]}{\sum_{i=1}^{n} \mathbb{I}[(h(\boldsymbol{x}_i)-y_i)^2 \geq c(\boldsymbol{x}_i)]}$.

### D.2   Some Results for Hinge Loss

In this section, we show some experimental results of the surrogate loss equipped with hinge loss, which can be formulated as follows:

$$\psi(h,r,c,\boldsymbol{x},y) = (h(\boldsymbol{x})-y)^2 \max(0, 1+r(\boldsymbol{x})) + c(\boldsymbol{x})\max(0, 1-r(\boldsymbol{x})).$$

Table 5 and Table 6 show some of the experimental results on the AgeDB adn BreastPathQ when surrogate loss equipped hinge loss, respectively. From this table, we can see that RcRLoss and AL is always lower than Sup in almost all experiments, which means that our method is effective in identifying test instances should be accepted and test instances should be rejected. It is worth noting that in most experiments, there is a low RA, which means that there is a higher tendency to reject hard-to-predict test instances to avoid serious errors when equipped hinge loss.

### D.3   Some Results for Logistic Loss

In this section, we show some experimental results of the surrogate loss equipped with logistic loss, which can be formulated as follows:

$$\psi(h,r,c,\boldsymbol{x},y) = (h(\boldsymbol{x})-y)^2 \log(1+\exp(r(\boldsymbol{x}))) + c(\boldsymbol{x})\log(1+\exp(-r(\boldsymbol{x}))).$$

Table 7, Table 8 and Table 9 show some of the experimental results on the UCI datasets and Breast-PathQ when surrogate loss equipped logistic loss, respectively. Our proposed method significantly outperforms the supervised regression method in almost all cases, which verifies the ability of our

**Table 11:** Test performance (mean and std) of our surrogate loss equipped square loss on five UCI datasets trained with the MLP model. We repeat the sampling-and-training process 10 times. The metrics Rej, AR, and RA are scaled to 0-100.

| Datasets | Cost | Supervised | Rej | AL | RL | Rej | AR | RA |
|---|---|---|---|---|---|---|---|---|
| Abalone | 3 |  | 2.39 (0.10) | 1.96 (0.19) | 7.82 (0.63) | 42.54 (2.49) | 32.79 (2.69) | 32.58 (2.63) |
|  | 4 | 4.44 (0.46) | 2.84 (0.16) | 2.33 (0.25) | 8.79 (0.97) | 31.82 (1.87) | 23.72 (2.14) | 40.81 (2.77) |
|  | 5 |  | 3.18 (0.18) | 2.60 (0.27) | 9.89 (1.15) | 25.37 (2.13) | 18.32 (2.07) | 45.65 (4.21) |
|  | 6 |  | 3.50 (0.29) | 2.89 (0.42) | 10.40 (1.11) | 20.38 (2.17) | 14.37 (1.85) | 49.83 (5.47) |
| Airfoil | 9 |  | 6.40 (0.25) | 4.36 (0.36) | 51.93 (5.13) | 43.65 (3.26) | 22.05 (2.93) | 22.22 (3.71) |
|  | 12 |  | 7.46 (0.31) | 5.11 (0.38) | 61.13 (5.83) | 33.75 (2.90) | 15.04 (3.10) | 27.05 (2.74) |
|  | 16 | 12.96 (2.60) | 8.57 (0.40) | 5.81 (0.30) | 70.20 (7.82) | 26.98 (3.23) | 10.54 (3.08) | 28.83 (1.79) |
|  | 20 |  | 9.27 (0.42) | 6.66 (0.43) | 76.90 (10.02) | 19.34 (2.34) | 7.70 (1.54) | 35.09 (4.65) |
|  | 25 |  | 9.97 (0.60) | 7.23 (0.51) | 87.37 (9.07) | 15.35 (2.44) | 5.19 (1.38) | 32.96 (5.61) |
|  | 30 |  | 10.33 (0.86) | 7.82 (0.79) | 85.67 (18.03) | 11.23 (1.95) | 3.92 (1.25) | 37.40 (8.27) |
| Auto-mpg | 4 |  | 3.65 (0.26) | 2.83 (0.90) | 11.93 (3.22) | 62.31 (11.46) | 51.76 (12.43) | 22.05 (10.30) |
|  | 6 |  | 5.19 (0.77) | 4.31 (1.47) | 18.00 (7.69) | 39.62 (21.51) | 33.51 (21.46) | 47.55 (24.43) |
|  | 8 | 8.34 (2.16) | 6.51 (1.35) | 5.82 (1.74) | 22.62 (8.96) | 29.10 (14.57) | 22.28 (13.86) | 52.29 (16.21) |
|  | 10 |  | 6.80 (1.16) | 6.08 (1.43) | 23.57 (9.41) | 17.82 (2.84) | 13.91 (1.93) | 65.62 (9.09) |
|  | 13 |  | 7.28 (1.30) | 6.45 (1.36) | 30.51 (15.46) | 12.69 (3.74) | 10.05 (3.46) | 71.16 (15.70) |
| Housing | 9 |  | 8.41 (1.56) | 8.22 (2.10) | 53.44 (20.25) | 28.22 (7.81) | 21.42 (7.35) | 56.77 (9.38) |
|  | 12 | 12.57 (3.43) | 9.03 (1.26) | 8.36 (1.64) | 76.10 (47.37) | 17.13 (5.71) | 12.16 (5.11) | 66.75 (11.99) |
|  | 16 |  | 8.52 (1.35) | 7.64 (1.62) | 109.61 (60.72) | 10.10 (3.67) | 7.30 (2.71) | 73.14 (13.36) |
|  | 20 |  | 9.40 (1.94) | 8.56 (2.16) | 148.19 (98.03) | 7.03 (3.30) | 5.09 (2.31) | 73.76 (16.54) |
| Concrete | 20 |  | 19.95 (2.56) | 18.77 (5.05) | 75.19 (11.68) | 55.10 (13.77) | 43.24 (14.20) | 28.28 (11.92) |
|  | 30 |  | 25.22 (3.22) | 22.44 (5.34) | 103.99 (18.06) | 33.45 (6.93) | 22.25 (6.21) | 43.75 (9.93) |
|  | 40 | 34.44 (3.05) | 31.21 (1.50) | 28.84 (1.84) | 127.77 (19.65) | 21.17 (5.11) | 12.47 (4.12) | 56.12 (7.44) |
|  | 50 |  | 29.55 (2.97) | 25.00 (3.80) | 147.99 (36.87) | 18.01 (4.62) | 9.87 (3.88) | 52.49 (9.18) |
|  | 60 |  | 33.07 (3.51) | 28.81 (3.94) | 158.65 (33.36) | 13.64 (3.71) | 7.28 (2.91) | 59.55 (8.52) |

method to reject difficult test instances demonstrating the effectiveness of our method. In most cases, the average loss of our method in the accepted test instances (AL) is always smaller than the average loss of the supervised regression model (Sup) in all test instances. This further indicates the ability of our method to identify hard-to-predict samples and reject them. On both MLP and Linear models, our method is effective in avoiding serious errors, which verifies that our method can be adapted to different models.

### D.4 Some Results for Square Loss

In this section, we show some experimental results of the surrogate loss equipped with square loss, which can be formulated as follows:

$$\psi(h, r, c, \boldsymbol{x}, y) = (h(\boldsymbol{x}) - y)^2 (r(\boldsymbol{x}) + 1)^2 + c(\boldsymbol{x})(r(\boldsymbol{x}) - 1)^2.$$

**Table 12:** Test performance (mean and std) of our surrogate loss equipped sigmoid on BreastPathQ. We repeat the sampling-and-training process 5 times. The metrics Rej, AR, RA are scaled to 0-100 and Sup, RcRLoss, AL and RL are all magnified by a factor of 1000.

| Cost | Sup | Rej | AL | RL | Rej | AR | RA |
|---|---|---|---|---|---|---|---|
| 5 | | 4.42 (0.17) | 2.40 (1.01) | 43.86 (10.60) | 79.22 (1.54) | 56.41 (4.19) | 2.81 (0.92) |
| 10 | | 8.44 (0.46) | 5.09 (1.64) | 51.35 (7.30) | 69.34 (2.56) | 47.12 (3.83) | 6.39 (1.83) |
| 15 | 16.77 (1.22) | 11.63 (0.38) | 6.30 (1.83) | 58.40 (14.79) | 60.23 (6.75) | 41.23 (5.42) | 10.88 (5.57) |
| 20 | | 14.31 (0.66) | 8.91 (1.26) | 57.83 (9.11) | 49.19 (4.68) | 33.84 (5.32) | 17.31 (3.13) |
| 25 | | 16.61 (0.60) | 9.09 (1.53) | 95.10 (59.42) | 47.38 (2.92) | 29.46 (4.84) | 17.78 (4.73) |

Table 10 and Table 11 show some of the experimental results on the BreastPathQ and UCI datasets with MLP model when surrogate loss equipped square loss, respectively. When the rejection cost $c$ is small, both RcRLoss and AL are significantly smaller than Sup. When the rejection cost $c$ is large, RcRLoss and AL are close to Sup but always smaller, which shows the effectiveness of our method to deal with regression with cost-based rejection.

### D.5 Some Results for Sigmoid

In this section, we show some experimental results of the Sigmoid function equipped with sigmoid, which can be formulated as follows:

$$\psi(h, r, c, \boldsymbol{x}, y) = (h(\boldsymbol{x}) - y)^2 \text{sigmoid}(r(\boldsymbol{x})) + c(\boldsymbol{x})\text{sigmoid}(-r(\boldsymbol{x})).$$

Unlike other binary classification losses, sigmoid can be viewed as weight balancing prediction loss and the rejection cost due to $\text{sigmoid}(r(\boldsymbol{x})) + \text{sigmoid}(-r(\boldsymbol{x})) = 1$. Table 12 show some of the experimental results on BreastPathQ when surrogate loss equipped sigmoid, respectively. RcRLoss and AL are always smaller than Sup, verifying the effectiveness of our method.

In our experiments, we used multiple binary classification losses (MAE, hinge loss, logistic loss, square loss and sigmoid) and different datasets including two deep datasets (BreastPathQ and AgeDB) and five uci datasets (Abalone, Airfoil, Auto-mpg, Housing and Concrete), and our method outperformed supervised regression in most cases, which demonstrates the effective of our method.

### D.6 Performance of Increasing Training Data

As we showed in Theorem 8 and Theorem 9, the pair $(h, r)$ learned by our surrogate loss could converge to the optimal pair $(h^*, r^*)$ when the number of training examples approaches to infinity. Therefore, the performance of the model can be improved as the training samples increase. To empirically validate such a theoretical finding, we further conduct experiments on the Abalone and Auto-mpg datasets by changing the fraction of training examples (100% means that we use all training examples in the training set). As shown in Figure 2, the RcRLoss and the AL generally decreases when more training examples are used for model training. This observation is clearly in accordance with our theoretical analyses in Theorem 8 and Theorem 9, because the learned model would be closer to the optimal model as more training examples are provided.

### D.7 Curves of AL and Rej

We show the AL loss for all methods with different rejection rates on Abalone, Autompg and Concrete datasets in Figure 3. In the plot of the relationship between AL loss and rejection rate, a curve at the bottom means better results.

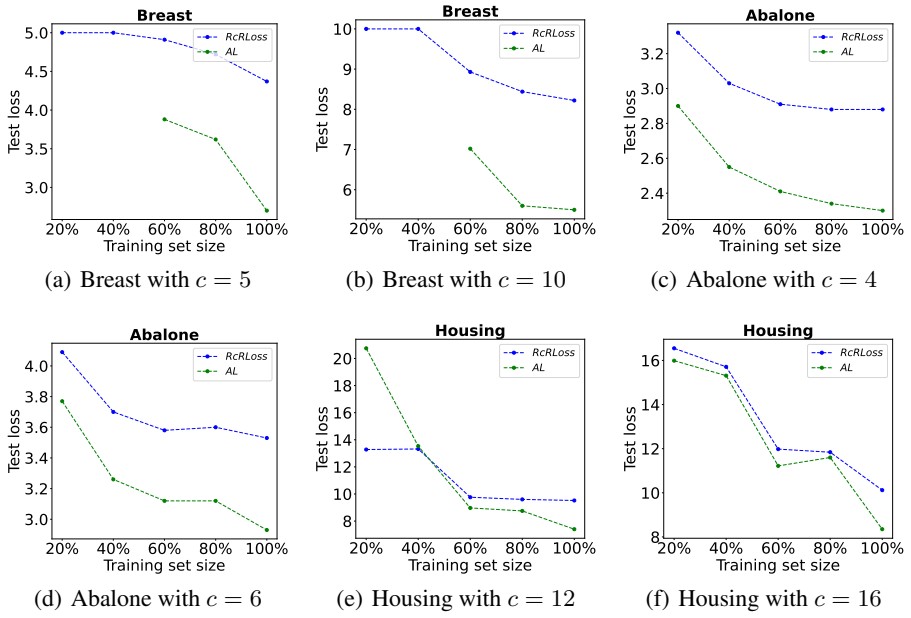

**Figure 2:** The test performance on the Breast, Abalone and Housing datasets for the surrogate loss equipped MAE when the number of training examples increases.

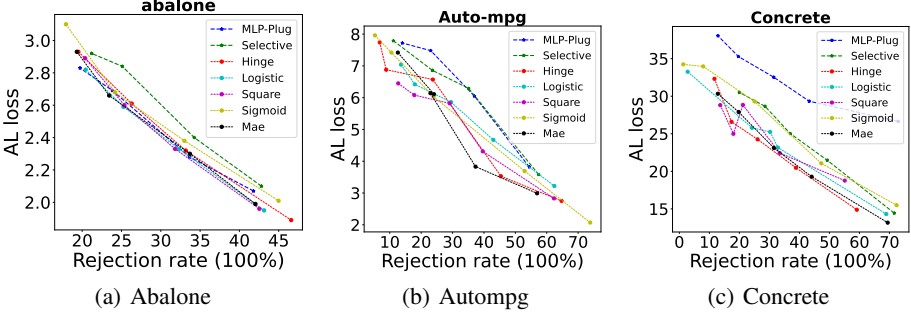

**Figure 3:** Figures (a), (b) and (c) report the accepted loss (AL) for all methods with different rejection rates on abalone, auto-mpg and concrete datasets, respectively.

# E    Limitations

In Theorem 4 and Theorem 5, we show that there is a limitation in our proposed method that requires the binary classification loss $\ell(r(\boldsymbol{x}), z)$ to be always greater than 0. This is easily satisfied by the design of the binary classification loss. However, to avoid the modification of the binary classification loss, we further propose Theorem 6, which only requires the binary classification loss to be non-negative, and this is easily satisfied. Extensive experiments on various datasets demonstrate the effectiveness of our proposed method. on the other hand, we use the pointwise cost functions $c(\boldsymbol{x})$ in our theoretical analysis, which shows that our method can be used for pointwise cost functions. For simplicity, in our experiments, we only consider the rejection cost as a constant $c$, but our method can easily handle the various pointwise cost functions in different application scenarios.

