# OpenReview forum: "Regression with Cost-based Rejection"
_NeurIPS.cc/2023/Conference — NeurIPS 2023 poster_

### Official Review · Reviewer_mRfi · 2023-07-02

**Soundness:** 3 good
**Presentation:** 3 good
**Contribution:** 4 excellent
**Rating:** 7
**Confidence:** 4

**Summary:**

Learning with rejection is an important machine learning problem. Most of the existing papers focus on the classification setting, i.e., classification with rejection and selective classification, and seldom works are targeting at the regression setting. This paper aims to investigate regression with cost-based rejection. Although some papers have studied the selective regression problem, I consider that the problem of regression with cost-based rejection is new.

To solve this new problem, this paper gives a formulation of the expected risk and derives the Bayes optimal solution. To train an ideal model, this paper also proposes a surrogate loss function that regards rejection as binary classification and provides conditions for the consistency. Experiments are conducted to demonstrate the effectiveness of the proposed.


**Strengths:**

- The problem of regression with cost-based rejection is interesting and new.

- It is quite important to give the formulation of expected risk for cost-based rejection and the Bayes optimal solution is meaningful and significant, which might serve as a pioneer for follow-up works to check whether the derived model and rejector are consistent when the mean squared error is used as the evaluation metric.

- A reasonable approach to training a good regression model with rejection is proposed and theoretical analyses are provided.

- Experimental results are significant, which supports the importance of considering cost-based rejection in the regression setting.


**Weaknesses:**

In Theorem 4, I notice that the authors did not introduce the concept "classification calibrated binary classification loss". For this concept, I also think that some references are required, e.g., [1] and [2].

[1] P. Bartlett et al. Convexity, classification, and risk bounds. JASA 2006.
[2] A. Tewari and P. Bartlett. On the consistency of multi-class classification methods. JMLR 2007.

- I also notice that the first two cited references are repeated. I would suggest that the authors should further check the details of the references.

- I also find some typos in this paper, e.g., missing a right parenthesis in Eq. (2).

- It will be interesting to give a general Bayes optimal solution for arbitrary regression losses, instead of limiting to the mean squared error.


**Questions:**

Compared with the selective setting, what is the key challenge of the proposed setting regression with cost-based rejection? The authors are encouraged to further explain this point.

---

> ### Author Rebuttal · Authors · 2023-08-10
>
> Thank you for your constructive review!
>
> **Q1: The concept of "classification calibrated binary classification loss" is missing.**
>
> A1: Thank you for pointing out this issue, which will be very helpful for us to improve our paper. We have rechecked our manuscript and added the concept to the revised manuscript.
>
> **Q2: It will be interesting to give a general Bayes optimal solution for arbitrary regression losses, instead of limiting to the mean squared error.**
>
> A2: We agree that it is interesting and meaningful to give a general Bayes optimal solution for arbitrary regression losses. Since the mean squared error (MSE) is the most widely used regression loss, we only considered the case of MSE in our paper. Besides, if we use another regression loss, the derivation process would be much different and the corresponding optimal solution may be extremely different from that of MSE. Therefore, whether there exists a general Bayes optimal solution is unknown, which is also a huge challenge to be solved in the future. We believe that the study of the Bayes optimal solution for other regression losses or the general Bayes optimal solution will be an interesting research direction for future work.
>
> **Q3: Compared with the selective setting, what is the key challenge of the proposed setting regression with cost-based rejection?**
>
> A3: As shown in Theorem 2 in our paper, compared with the selective setting, the key challenge of regression with cost-based rejection is how to accurately estimate the variance of the real-valued label $y$ over the distribution $p(y|x)$. However, estimating the variance without knowing the distribution $p(y|x)$ is difficult. Unfortunately, empirically estimating $p(y|x)$ is also difficult because $p(y|x)$ is unknown and needs to be approximated by a well-performed model.
>
> **Q4: Miscellaneous minor issues.**
>
> A4: Thank you for pointing out these issues, which will be very helpful for us to improve our paper. We have rechecked our manuscript and corrected the clarity issues in the revised manuscript. For your mentioned issues, we have removed the repeated references. Besides, we have added the missing parentheses to Eq. (2).

---

> > ### Comment · Reviewer_mRfi · 2023-08-11
> > **To response**
> >
> > Thank you for your answers. My concerns have been addressed.

---

> > > ### Author Response · Authors · 2023-08-15
> > > **Thank you for supporting our paper**
> > >
> > > Thank you for your reply! We also sincerely thank you for your valuable time on our paper and thank you for supporting our paper.

---

### Official Review · Reviewer_J2H6 · 2023-07-03

**Soundness:** 3 good
**Presentation:** 3 good
**Contribution:** 3 good
**Rating:** 6
**Confidence:** 4

**Summary:**

The paper addresses problem of regression with the reject option. The authors propose the cost-based formulation of an optimal reject-option regression rule and they derive (Bayes) optimal strategy for the case the distribution is known. The authors further propose a surrogate loss to learn the reject-option regression rule from examples. They prove the the consistency and regret bounds for the proposed learning approach.

**Strengths:**

The paper is sound and it is very clearly written.

The proposed method based on surrogate loss is simple and potentially effective.

The authors derive theoretical guarantees for the proposed estimator, namely, they show the consistency and the regret bound.

**Weaknesses:**

The first contribution, i.e. formulation of the cost-based reject option regression and the optimal solution, is a known result. E.g. it is given in [41], see equation (1) of the paper. Besides [41], deriving the optimal strategy for a generic reject option predictor (of which the regression with L2-loss is a special case) is straightforward and appears in pattern recognition textbooks, e.g. Schlesinger et al. Ten Lectures on Statistical and Structural Pattern Recognition. Springer 2002.

The proposed method, based on minimizing the surrogate loss, is not compared against any baseline solution nor any existing methods for learning reject option regression. As a result, when there is no reference, it is difficult to judge about efficiency of the proposed method. The minimal solution would be to use synthetic data with known ground-truth. On real data one could use any regression model which outputs estimate p(y|x), like e.g. Bayesian methods, and plugin Bayes rule.  The author may argue that most existing methods formulate the optimal reject-option regression using the concept of selective risk and coverage [41][20][38]. However, all methods (including the proposed approach) can be compared in terms of the selective risk and the coverage which are reported by the authors anyway in the experiments (section 5) although the authors use different terminology. Namely, the selective risk is denoted as the "accepted loss" AL and the coverage equals 1 - rejection rate (RR). Note the cost-based formulation and the selective risk vs. coverage formulation (known as the bounded-improvement or bounded-abstention rejection models) are equivalent in the sense that both lead to the same Bayes-optimal solution, i.e. setting the rejection cost (as in the paper under review) has the same effect as setting threshold on the coverage (or the selective risk), see e.g. Franc et al. Optimal strategies for reject option classifiers. JMLR 2023.

Minor problems:

- Regarding the experiments in sec 5, errors observed on AgeDB dataset are excessively large. The mean error ~100, reported for the standard regression model (sup), makes no sense for age prediction regardless whether the authors report MAE or L2-loss which is not clear from the description.

- The observations derived from the experiments (section 5.5) are questionable or trivial: "(1) Our proposed method significantly outperforms the supervised reression method" It is not clear in what sense the propsed method is better as it solves a different problem than the non-reject model. "(2) In most cases, the average loss of our method in the accepted test instances (AL) is always smaller than the average loss of the supervised regression model"; note that this holds true for any rejection rule regardless how good the rejector $r(x)$ is. Similarly the obsevation (3) is obvious and it hold for any rejection rule.

- Line 273: "...RcR loss (RcRLoss) decreases" -> increases

**Questions:**

Please explain reasons for not using any baseline method in the experimental evaluation?

---
The authors satisfactorily addressed my questions in the rebuttal based on which I increased my ratings.

**Limitations:**

yes

---

> ### Author Rebuttal · Authors · 2023-08-10
>
> Thank you so much for your valuable comments!
>
> **Q1: The first contribution regarding the formulation of the cost-based reject option regression and the optimal solution.**
>
> A1: We agree that our derived optimal solution of regression with cost-based rejection is quite similar to the result in [41], but we would like to humbly argue that our derived optimal solution is more general and suitable for regression with cost-based rejection instead of selective regression. In [41], equation (1) is derived by introducing the tuning parameter $\lambda$ which is responsible for compromising error and rejection rate. Hence $\lambda$ can only be taken as a constant rejection cost. In contrast, our formulation is more general and can be naturally compatible with instance-dependent rejection cost, and thus is more suitable for regression with cost-based rejection. Therefore, despite being quite similar to the result in [41], we believe that our formulation is also meaningful. We admit that the derivation process of such an optimal solution with L2-loss is straightforward, which is also similar to that of [41]. However, since regression with cost-based rejection is a new problem that has not been studied before, it is quite essential to explicitly show the general optimal solution with instance-dependent cost. Thank you for providing us with such an insightful comment and we will properly revise our manuscript according to the above constructive discussionwith you.
>
> [41] A. Zaoui, C. Denis, and M. Hebiri. Regression with reject option and application to knn. In NeurIPS, 2020.
>
> **Q2: Comparison with selective regression.**
>
> A2: Since this question was repeatedly mentioned by reviewers, we think it is very important. We provide a detailed response to this question in Global Response, so please refer to **Global Response** to check the full response. Briefly, since our paper provides the first attempt to investigate regression with cost-based rejection, there does not exist a baseline that can be directly compared, so we propose a number of evaluation metrics and conduct extensive experiments to verify the validity of our method. However, this does not seem to be convincing enough, so we have adopted the suggestions of reviewers to add comparisons with the methods in selective regression [19, 41] to make our experimental results more convincing. Based on the comparison of accepted loss (AL), our proposed method outperforms (smaller AL with the same rejection rate) all compared methods, which validates the effectiveness of our method.
>
> [19] Y. Geifman and R. El-Yaniv. Selective classification for deep neural networks. In NeurIPS, 2017.
>
> [41] A. Zaoui, C. Denis, and M. Hebiri. Regression with reject option and application to knn. In NeurIPS, 2020.
>
> **Q3: The cost-based formulation and the selective risk vs. coverage formulation (known as the bounded-improvement or bounded-abstention rejection models) are equivalent in the sense that both lead to the same Bayes-optimal solution.**
>
> A3: Thank you so much for pointing out this conclusion by Franc et al. (2023) that the cost-based and sensitivity (known as the bounded-improvement or bounded-abstention rejection models) frameworks have the same Bayes-optimal solution. The conclusion can help us to theoretically connect different rejection-based frameworks. However, even if different frameworks have the same optimal solution, they actually do not face the same challenges and application scenarios. Selective regression aims to reject with a fixed rejection rate, while regression with cost-based rejection aims to reject with a fixed rejection cost. To the best of our knowledge, we provide the first study on regression with cost-based rejection. We cannot expect the first study (especially the conference paper) to accomplish all relevant tasks, although it is interesting and meaningful to analyze the connections with different rejection frameworks. Therefore, we think it is more appropriate to leave the study of theoretical links between different rejection frameworks for future work.
>
> **Q4: Excessive errors observed on the AgeDB dataset.**
>
> A4: Thank you for pointing out the lack of clarity in the presentation of our table. Specifically, all of our results are based on L2-loss (i.e., Mean Squared Error). We admit that an average error of about 100 even for the standard regression model (sup) on the AgeDB dataset is excessive. In fact, these results are normal without the use of additional training data, and similar errors also have appeared in a previous paper, see e.g. Yang et al. Delving into deep imbalanced regression. ICML 2021.
>
> **Q5: The observations derived from the experiments (section 5.5) are questionable or trivial.**
>
> A5: Thank you for pointing out this issue. We agree with you that these observations seem somewhat trivial because it may not be appropriate to directly compare our work with supervised regression. Our original intention is to show the effectiveness of our proposed method in regression with cost-based rejection. We will properly revise our manuscript for clarifying this part.
>
>
> **Q6: Please explain the reasons for not using any baseline method in the experimental evaluation?**
>
> A6: To the best of our knowledge, our paper provides the first attempt to investigate regression with cost-based rejection (RcR), hence there are no baseline methods for this task. In order to show the effectiveness of our method, we propose a number of evaluation metrics and conduct extensive experiments. However, these results do not seem convincing enough, so we added the comparison experiments with the selective regression method in Global Response. Please refer to **Global Response** to check the comparison experiments.

---

> > ### Comment · Reviewer_J2H6 · 2023-08-15
> > **response**
> >
> > Thank you for your elaborated response to my review. Please allow me a response to your response.
> >
> > add Q1/A1:
> > I agree that your formulation differs from the known setup by having a different reject-cost per instance. Except this modification the setup and the solution is the same as in [41]. In the experiments you use a single reject-cost which does not demonstrate that the modification is acutely needed, however, you may have an application in mind where the instance specific cost is need ? I am glad that you agree that deriving the Bayes strategy is a straightforward. It was surprising for me that this tiny modification is claimed as the first main contribution of the paper.
> >
> > add Q2/A2:
> > Thanks for additional experiments. Please describe how did you construct the uncertainty score for method [19]. Did you use MC-dropout as the uncertainty score?
> >
> > I would suggest slightly different way to compare the methods. Take any baseline approach which outputs an uncertainty score for given input and construct the risk-coverage curve (x-axis is the reject-rate; y-axis is the accepted loss using you terminology). Run you method for fixed reject-cost(s) and draw the obtained pair (rejection-rate, accepted loss) as a point to the risk-coverage curve. If the point(s) are below the risk-coverage curve of the baseline your method wins. As a baseline I would propose e.g. MC dropout as in [19].
> >
> > add Q3/A3 and Q6/A6:
> > The point of this comment is that the key problem when constructing reject option rule is the estimation of the uncertainty score. The optimal strategy for various formulation (e.g. cost-based formulation as in your paper or bounded-improvement as in "selective-classification" papers) is always based on thresholding uncertainty score (note that the optimal strategy you derived is also based on thresholded conditional variance). Once you have the uncertainty score, the threshold can be easily found on validation set based on the criterion you want, e.g. using sample estimate of (10). In turn any existing method on selective regression which outputs an uncertainty score can be readily applied to the cost-based problem you investigate in your paper. Therefore it is not difficult to construct a baseline for you method.

---

> > > ### Author Response · Authors · 2023-08-18
> > > **Thank you for your reply**
> > >
> > > Thank you for your insightful comment.
> > >
> > > **Add Q1/A1:**
> > >
> > > We indeed have some applications in mind where the instance-specific rejection cost is required. For example, in safety-critical applications such as autonomous driving, we need to predict the suitable deflection angle for the tires to pass through an intersection based on sensor information. Obviously, the difficulty and risk faced by an intersection that can only be passed with a deflection angle from $5^{\circ}-10^{\circ}$ is not the same as an intersection that can be passed with a deflection angle from $10^{\circ}-40^{\circ}$, therefore they do not have the same rejection cost (rejection cost $2.5^{\circ}$ in the first intersection and rejection cost $15^{\circ}$ in the second intersection).
> > >
> > > We will not take Theorem 2 as one of the main contributions of our paper. In this case, our main contributions in this paper are summarized as follows:
> > >
> > > - We propose a surrogate loss function considering rejection as a binary classification process and give a condition of regressor-consistent that the classification calibrated binary classification loss is always greater than 0. In that condition, the optimal regressor can be derived by our method.
> > >
> > > - We propose a definition of rejector-calibration and show that our method is rejector-calibration when the regressor-consistent condition is satisfied. Based on this, we further propose a weaker version of the condition allowing the classification calibrated binary classification loss to be greater than or equal to 0. In the weakened condition, the regression consistency can only be satisfied in the accepted instances, and regressor-consistent is still satisfied.
> > >
> > > - We derive the theoretical analysis of the regret transfer and estimation error bounds for our proposed method, and extensive experiments demonstrate the effectiveness of our method.
> > >
> > >
> > > **Add Q2/A2:**
> > >
> > > We are really sorry that we have mistakenly linked [19] to the paper of [18], in our previous response. The correct links (in accordance with our originally submitted main paper) are as follows:
> > >
> > > [18] Y. Geifman and R. El-Yaniv. Selective classification for deep neural networks. In NeurIPS, 2017.
> > >
> > > [19] Y. Geifman and R. El-Yaniv. Selectivenet: A deep neural network with an integrated reject option. In ICML, 2019.
> > >
> > > This mistake is because we removed duplicate references in our revised version (not uploaded yet) and thus the serial numbers of the original references were changed.
> > >
> > > We really appreciate that you provided a wonderful visualized way (risk-coverage curve) to compare different methods for regression with rejection. Unfortunately, we cannot add images to OpenReview in the current stage, but we promise that we will definitely include these visualized figures in our final version.
> > >
> > > **Add Q3/A3 and Q6/A6:**
> > >
> > > We agree that the estimation of the uncertainty score is quite important to regression with rejection, which can be used as the metric to determine whether an example will be rejected. We also agree that the uncertainty score (the conditional variance) can be used to solve the problem we studied in this paper and such a method can be considered as a baseline. One intuitive baseline is to use the Plug technique proposed in [41], which we have already compared with (i.e., **MLP-Plug**). We will be much appreciated if you have other suitable methods to estimate the conditional variance and we will definitely try our best to include them for comparisons.
> > >
> > > In fact, we believe that our proposed method is better than those baselines relying on the uncertainty score. By Occam's razor principle, calculating the uncertainty score is an intermediate, which is normally not the optimal solution, especially when the uncertainty score cannot be accurately estimated. In contrast, our proposed method gives a direct solution by taking rejection as binary classification. Therefore, our method is expected to achieve better performance.
> > >
> > >
> > > [41] A. Zaoui, C. Denis, and M. Hebiri. Regression with reject option and application to knn. In NeurIPS, 2020.

---

> > > > ### Comment · Reviewer_J2H6 · 2023-08-20
> > > >
> > > > I appreciate the authors' effort and how they addressed all my concerns. I am satisfied with the replies. I will rise my scoring.

---

> > > > > ### Author Response · Authors · 2023-08-20
> > > > >
> > > > > Thank you so much for your reply! Your insightful comments really helped us a lot to improve our paper (especially on the comparative experiments with existing methods). We will definitely incorporate the above constructive discussions with you into our revised version. We sincerely appreciate your informative feedback and valuable time on our paper!

---

### Official Review · Reviewer_4Fnw · 2023-07-04

**Soundness:** 3 good
**Presentation:** 3 good
**Contribution:** 3 good
**Rating:** 5
**Confidence:** 3

**Summary:**

This paper focuses on the problem of regression with rejection, specifically the approach of specifing a cost function and learn the pair of regressor and rejector at the same time. The paper prensents a concrete path to solving the problem. It first properly defines the problem and shows the Bayes optimal solution to it. Since the Bayes optimal solution requires knowing the expectation and the variance of the underlying distribution, it then proposes a learnable risk defined using surrogate loss function. Then, the paper theoretically investigate and show the usefullness of the proposed approach, from the perspective of classification calibration and error bound. Finally it empirically evaluates the proposed method on several typical datsets using various metrics.


**Strengths:**

Originality:
This paper tackles the regression with rejection problem which is of significant importance in the field. The paper solves the problem from a novel perspective and can be seen as a novel combination of several well-known techniques. This paper addresses clearly how it is related and different from related publications.

Quality:
The paper is technically sound and self-contained. Its claims are properly supported by theoretical demonstrations.

Clarity:
The paper is clearly written and easy to follow. The structual is well organized.

Significance:
The proposed method has significance to some extent, as it considers a new approach to an important problem.

**Weaknesses:**

- Empricail comparison is no sufficiently conducted.
  - There is no comparison with existing methods.
  - There is no investigation on varying cost.
  - There is no investigation on slow-start. This would show the robustness of the proposed method, since slow-start introduces a hyper-parameter to the method.
  - There is no investigation on varying training data size. Some theoretical results shows how performance would change on different $n$. It would be pursuative to show it empirically to some extent.


**Questions:**

- For the cost funciton $c(x)$, it is used as a function on $x$ instead of a constant in theoretical demonstrations but considered as a constant in experiments. Does theoretical results has some relationship or limitation on the form of the pointwise cost function? Does some results rely on cost being a non-constant function?

- Are there any detailed discussion on the slow-start mechanism, since is part is not covered by any theory but has crucial practical importance? For example, how the slow-start epochs affect the overall performance? Can we completely stop the learning of $h$ after the slow-start epochs?


**Limitations:**

Techinal limitations on loss function are addressed by authors in appendix.

---

> ### Author Rebuttal · Authors · 2023-08-10
>
> Thank you so much for your valuable comments!
>
> **Q1: There is no comparison with existing methods.**
>
> A1: Since this question was repeatedly mentioned by reviewers, we think it is very important. We provide a detailed response to this question in Global Response, so please refer to **Global Response** to check the full response. Briefly, since our paper provides the first attempt to investigate regression with cost-based rejection, there does not exist a baseline that can be directly compared, so we propose a number of evaluation metrics and conduct extensive experiments to verify the validity of our method. However, this does not seem to be convincing enough, so we have adopted the suggestions of reviewers to add comparisons with the methods in selective regression [19, 41] to make our experimental results more convincing. Based on the comparison of accepted loss (AL), our proposed method outperforms (smaller AL with the same rejection rate) all compared methods, which validates the effectiveness of our method.
>
> [19] Y. Geifman and R. El-Yaniv. Selective classification for deep neural networks. In NeurIPS, 2017.
>
> [41] A. Zaoui, C. Denis, and M. Hebiri. Regression with reject option and application to knn. In NeurIPS, 2020.
>
> **Q2: There is no investigation on varying costs.**
>
> A2: We would like to kindly remind that we have already set various rejection costs $c$ for each dataset and performed repetitive experiments. As the rejection cost $c$ increases, the RcR loss and accepted loss gradually increase while the rejection rate gradually decreases.
>
> **Q3: There is no investigation on slow-start. Are there any detailed discussions on the slow-start mechanism? How do the slow-start epochs affect the overall performance? Can we completely stop the learning of $h$ after the slow-start epochs?**
>
>
> A3: In the implementation details of our experiments, we use a training trick called slow-start, where we first only train the regressor $h$ and then train the regressor $h$ and the rejector $r$ together. The slow-start trick is to alleviate the detraining issue in the early training stage when $h$ is disturbed by $r$, especially when deep neural networks with gradient descent optimization are used. Similar tricks were also used in other rejection learning methods [19]. After our preliminary study, the hyper-parameter of slow-start epochs has no significant effect on the experimental results, and we suggest setting it to 20\% of the total training epochs. It is worth noting that we cannot stop training $h$ after slow-start because slow-start is only used to accelerate the training of $h$ in the early training stage.
>
> [19] Y. Geifman and R. El-Yaniv. Selective classification for deep neural networks. In NeurIPS, 2017.
>
> **Q4: There is no investigation on varying training data size.**
>
> A4: Thank you for the wonderful suggestion. We have added additional experiments by varying the sizes of training data. Specifically, we considered using 20\%, 40\%, 60\%, 80\%, 100\% of the training set to train our model. The experimental results are reported following table. As can be seen from following table, there is a significant reduction in RcR loss, AL, and Rej, as more training data is used. This is clearly in accordance with our intuition that the performance will be improved if more training data is provided.
>
> | Datasets | Cost | Slice | RR | AL | RL | Rej |
> | :----:| :----: | :----: | :----: | :----: | :----: | :----: |
> | Abalone | 3.0 | 20\% | 2.81 $\pm$ 0.16 | 2.15 $\pm$ 0.55 | 6.51 $\pm$ 1.30 | 72.31 $\pm$ 11.12 |
> | Abalone | 3.0 | 40\% | 2.60 $\pm$ 0.14 | 2.18 $\pm$ 0.29 | 8.28 $\pm$ 1.35 | 51.58 $\pm$ 10.11 |
> | Abalone | 3.0 | 60\% | 2.48 $\pm$ 0.14 | 2.05 $\pm$ 0.30 | 8.50 $\pm$ 1.09 | 46.74 $\pm$ 2.96 |
> | Abalone | 3.0 | 80\% | 2.45 $\pm$ 0.13 | 2.03 $\pm$ 0.24 | 8.22 $\pm$ 1.16 | 45.66 $\pm$ 3.83 |
> | Abalone | 3.0 | 100\% | 2.41 $\pm$ 0.12 | 1.99 $\pm$ 0.21 | 8.13 $\pm$ 1.08 | 42.04 $\pm$ 3.18 |
> | Auto-mpg | 4.0 | 20\% | 4.14 $\pm$ 0.17 | 4.83 $\pm$ 1.37 | 12.35 $\pm$ 2.85 | 82.37 $\pm$ 10.51 |
> | Auto-mpg | 4.0 | 40\% | 3.91 $\pm$ 0.27 | 3.19 $\pm$ 1.28 | 10.73 $\pm$ 2.96 | 73.97 $\pm$ 8.43 |
> | Auto-mpg | 4.0 | 60\% | 4.15 $\pm$ 0.78 | 3.68 $\pm$ 2.15 | 11.97 $\pm$ 2.68 | 69.10 $\pm$ 12.92 |
> | Auto-mpg | 4.0 | 80\% | 3.90 $\pm$ 0.75 | 3.47 $\pm$ 1.69 | 13.08 $\pm$ 3.22 | 60.00 $\pm$ 20.23 |
> | Auto-mpg | 4.0 | 100\% | 3.64 $\pm$ 0.29 | 2.99 $\pm$ 0.83 | 13.98 $\pm$ 4.16 | 56.92 $\pm$ 13.00 |
>
> **Q5: Does theoretical results has some relationship or limitation on the form of the pointwise cost function? Does some results rely on cost being a non-constant function?**
>
> A5: In our theoretical analysis, we use pointwise cost functions $c(x)$, which shows that our method can be used for pointwise cost functions. For simplicity, in our experiments, we only consider the rejection cost as a constant $c$, but our method can easily handle various pointwise cost functions in different application scenarios.

---

> > ### Comment · Reviewer_4Fnw · 2023-08-14
> > **Thanks authors and I remain positive**
> >
> > Thank you for detailed response and I really appreciate the effort for addressing concerns of comparison with related existing methods.
> > My concerns over slow-start trick and pointwise cost function are also addressed.
> > My concerns of empirical performance over varying data size is also clearly addressed using different datasets and costs.

---

> > > ### Author Response · Authors · 2023-08-15
> > > **Thank you for remaining positive**
> > >
> > > Thank you for letting us know that your concerns were addressed. We also sincerely thank you for your valuable time on our paper and thank you for remaining positive!

---

### Official Review · Reviewer_1UF6 · 2023-07-07

**Soundness:** 3 good
**Presentation:** 3 good
**Contribution:** 2 fair
**Rating:** 5
**Confidence:** 4

**Summary:**

The paper explores the framework of regression with rejection, in which the model can opt to refrain from making predictions on certain instances at specific costs, with the intention of avoiding critical mispredictions. The paper determines the Bayes optimal solution and introduces a theoretically grounded surrogate loss within the framework.

**Strengths:**

1. The paper is pioneering in studying the regression with rejection setting.
2. The presentation of the paper is clear and concise.

**Weaknesses:**

1. While the regression with rejection setting represents a fresh concept in the literature, the technical approach seems to closely mirror the standard classification with rejection setting. This resemblance potentially limits the novelty of the paper. Could the authors elaborate on the specific technical challenges encountered within this setting?

2. A definition of regressor-consistency that parallels rejector-calibration (Definition 3) is missing.

3. The use of a supervised regression method may not serve as an appropriate and fair baseline for rejection experiments. It would be more convincing to conduct experiments comparing some straightforward rejection methods against the proposed rejection methods.

4. Additional commentary on the experimental results is required. For instance, the setup considers a range of binary classification loss functions; which among these yields the best results based on the experiments conducted?



**Questions:**

See Weakness.

**Limitations:**

N/A.

---

> ### Author Rebuttal · Authors · 2023-08-10
>
> Thank you for your constructive review!
>
> **Q1: While the regression with rejection setting represents a fresh concept in the literature, the technical approach seems to closely mirror the standard classification with rejection setting. This resemblance potentially limits the novelty of the paper. Could the authors elaborate on the specific technical challenges encountered within this setting?**
>
> A1: Thank you for posing this nice question. We agree that our studied setting of regression with rejection shares some similarities with the setting of classification with rejection (CwR) due to the rejection option, but they are basically targeting two different tasks, i.e., regression and classification. The label space is continuous and infinite in regression while is discrete and finite in classification, which makes the labeling information harder to capture in regression than in classification. As shown in Theorem 2 in our paper, the main technical challenge of regression with cost-based rejection is **how to accurately estimate the variance of the real-valued label $y$ over the distribution $p(y|x)$**. However, estimating the variance without knowing the distribution $p(y|x)$ is difficult, and empirically estimating $p(y|x)$ is also difficult because $p(y|x)$ is unknown and needs to be approximated by a well-performed model. To address these challenges, we propose to a surrogate loss function considering rejection as a binary classification problem and provide theoretical analyses on regressor-consistency and rejector-calibration. There is an evident difference in the technical approach between our work and most of the existing studies on CwR, i.e., our proposed method avoids estimating $p(y|x)$ by taking rejection as binary classification while most of the existing studies on CwR need to accurately estimate $p(y|x)$ for rejection.
>
> **Q2: A definition of regressor-consistency that parallels rejector-calibration is missing.**
>
> A2: Thank you for pointing out this issue. A informal definition of regressor-consistency was provided in Lines 156-157 of our paper, i.e., we say a method is regressor-consistent, meaning that the regressor $h$ learned by the method converges to the optimal regressor $h^{*}$ as the number of training data increases. We will add a formal definition to our revised manuscript.
>
> **Q3: The use of a supervised regression method may not serve as an appropriate and fair baseline for rejection experiments. It would be more convincing to conduct experiments comparing some straightforward rejection methods against the proposed rejection methods.**
>
> A3: Since this question was repeatedly mentioned by reviewers, we think it is very important. We provide a detailed response to this question in Global Response, so please refer to **Global Response** to check the full response. Briefly, since our paper provides the first attempt to investigate regression with cost-based rejection, there does not exist a baseline that can be directly compared, so we propose a number of evaluation metrics and conduct extensive experiments to verify the validity of our method. However, this does not seem to be convincing enough, so we have adopted the suggestions of reviewers to add comparisons with the methods in selective regression [19, 41] to make our experimental results more convincing. Based on the comparison of accepted loss (AL), our proposed method outperforms (smaller AL with the same rejection rate) all compared methods, which validates the effectiveness of our method.
>
> [19] Y. Geifman and R. El-Yaniv. Selective classification for deep neural networks. In NeurIPS, 2017.
>
> [41] A. Zaoui, C. Denis, and M. Hebiri. Regression with reject option and application to knn. In NeurIPS, 2020.
>
> **Q4: Additional commentary on the experimental results is required. For instance, the setup considers a range of binary classification loss functions; which among these yields the best results based on the experiments conducted?**
>
> A4: Indeed, we found that different binary classification loss functions (e.g. mean absolute error, mean square loss, logistic loss, sigmoid, and hinge loss) perform differently on different datasets. For example, hinge loss and sigmoid have similar performances on the BreastPathQ dataset, but sigmoid loss performs better than hinge loss on the AgeDB dataset. Our experimental results show that the performance of MAE is more stable (no excessively high or low rejection rate), compared with other loss functions.

---

### Author Rebuttal · Authors · 2023-08-10

## Global Response

We sincerely appreciate the thoughtful feedback and insightful comments from all reviewers to help improve our work. We are glad that our study was recognized by the reviewers (Reviewer 1UF6, Reviewer 4Fnw, Reviewer J2H6 and Reviewer mRfi). We are delighted that they found our work to be novel (Reviewer 4Fnw), interesting (Reviewer mRfi), clear (Reviewer 1UF6), and simple (Reviewer J2H6), and we are encouraged by these positive comments.

We have corrected all the typos mentioned by reviewers and made the following key changes to our revised manuscript:

[Section 1] We made appropriate modifications to our first contribution regarding the formulation of the cost-based reject option regression and the optimal solution (Reviewer J2H6).

[Section 4.1] We added the definition of regressor-consistency (Reviewer 1UF6) and the concept of class-calibrated binary classification loss (Reviewer mRfi).

[Section 5] We added a comparison with the method of selective regression to demonstrate the validity of our method to make our work more convincing (Reviewer 1UF6, Reviewer 4Fnw and Reviewer J2H6).

[Section 5.5] We revised Section 5.5 to clearly express the validity of our method (Reviewer J2H6).

[Appendix D] We added experiments on varying training data sizes (Reviewer 4Fnw).

[Appendix E] We added a discussion of pointwise cost functions and constant costs (Reviewer 4Fnw).

----------

Below are our responses to the commonest concern regarding the experimental comparison with previous (selective regression) methods.

**Q: The experimental comparison with previous (selective regression) methods.**

A: Thank you for raising this concern, which will be definitely helpful for us to improve our paper! To the best of our knowledge, our paper provides the first attempt to investigate regression with cost-based rejection (RcR), hence there are currently no direct baseline methods for this problem. So we propose a number of evaluation metrics and conduct extensive experiments hoping to show the effectiveness of our method.

We admit that there are no directly comparable methods for our setting, but we can adopt the suggestions of reviewers to add comparisons with the methods in selective regression. We have conducted additional experiments to compare with SelectiveNet [19] and Knn-Plug [41], and the results are reported in Table 1 and Table 2, respectively (in the attached one-page PDF). Specifically, to be able to establish a connection between our studied RcR and selective regression, we set the expected rejection rate (RJ) of the selective regression method based on the results of RcR\_MAE (our surrogate loss equipped MAE). It is important to note that there is no way to perfectly match the rejection rate, as the set rejection rate (RJ) is "expected". From Table 1 and Table 2, our proposed method outperforms (smaller AL with the same rejection rate) all compared methods, which validates the effectiveness of our method.


Although RcR and selective regression seem to have a lot of similarities, they actually do not face the same challenges and application scenarios. The selective regression aims to reject with a fixed rejection rate, while RcR aims to reject with a fixed rejection cost. In many real-world scenarios, the rejection rate is inaccessible and we can only have the rejection cost. For example, in safety-critical applications such as autonomous driving, we need to predict the most suitable deflection angle $20^{\circ}$ for the tires based on sensor information. However it is not realistic to have a perfect prediction, so we have a tolerance angle of $10^{\circ}-30^{\circ}$ to complete the turn. In such a case, we can easily set a rejection cost $(10^{\circ})$ instead of a rejection rate. Based on such a view, we believe that RcR is worth studying.


[19] Y. Geifman and R. El-Yaniv. Selective classification for deep neural networks. In NeurIPS, 2017.

[41] A. Zaoui, C. Denis, and M. Hebiri. Regression with reject option and application to knn. In NeurIPS, 2020.

---

> ### Comment · Reviewer_J2H6 · 2023-08-15
> **What was the uncertainty score?**
>
> Thank you for additional experiments. The work [19] proposes an algorithm to tune a decision threshold of reject option classifier in order to guarantee a specific rejection-rate on test set. [19] does not propose any uncertainty score that would be applicable for reject option regression. Could you please explain which uncertainty score was used?
>
> Regarding [41], their metod uses kNN regression as the base model. Did you adapted their method to MLP based regression MLP?

---

> > ### Author Response · Authors · 2023-08-18
> > **More clarifications**
> >
> > We are really sorry that we have mistakenly linked [19] to the paper of [18], in our previous response. The correct links (in accordance with our originally submitted main paper) are as follows:
> >
> > [18] Y. Geifman and R. El-Yaniv. Selective classification for deep neural networks. In NeurIPS, 2017.
> >
> > [19] Y. Geifman and R. El-Yaniv. Selectivenet: A deep neural network with an integrated reject option. In ICML, 2019.
> >
> > This mistake is because we removed duplicate references in our revised version (not uploaded yet) and thus the serial numbers of the original references were changed.
> >
> > -------------------------
> > Let us introduce more implementation details for the comparative experiments with selective regression methods [19, 41].
> >
> > [19] Y. Geifman and R. El-Yaniv. Selectivenet: A deep neural network with an integrated reject option. In ICML, 2019.
> >
> > [41] A. Zaoui, C. Denis, and M. Hebiri. Regression with reject option and application to knn. In NeurIPS, 2020.
> >
> > For SelectiveNet [19], this work proposes a neural network architecture and an optimization goal to control the rejection threshold of the rejector to ensure a specific rejection rate. To ensure a fair comparison, we use the same base model for all datasets. The experimental results (in Table 1 of our attached one-page PDF) show that our method outperforms SelectiveNet in nearly all cases.
> >
> > For Knn-Plug [41], this method proposes a semi-supervised learning process for learning a data-driven predictor with a reject option based on the plug-in principle. Specifically, this method learns a regression function $h$ and a conditional variance function $\sigma$ from the labeled dataset, while the unlabeled dataset is used to calibrate the conditional variance function $\sigma$ to ensure the desired rejection rate. **In the previous experimental results in Table 2 (in our attached one-page PDF), we used the KNN regression as the base model.** For each dataset, we randomly split the original dataset into a labeled training set, an unlabeled training set, and a test set by the proportions of $60\%$ (the same size of labeled training set as ours), $20\%$, and $20\%$, respectively. **We further provide additional experimental results using the MLP model as the base model, in the following table.** As can be seen from this table, our method outperforms MLP-Plug in nearly all cases.
> >
> > | Datasets | Cost | RcR\_MAE AL | RcR\_MAE Rej | rj | MLP-Plug AL | MLP-Plug Rej |
> > | :----:| :----: | :----: | :----: | :----: | :----: | :----: |
> > | Airfoil | 9 | **4.23 $\pm$ 0.86** | 62.23 $\pm$ 3.73 | 62.23 | 5.37 $\pm$ 0.93 | 63.26 $\pm$ 4.69 |
> > |  | 12 | **5.39 $\pm$ 0.86** | 40.33 $\pm$ 7.95 | 40.33 | 7.58 $\pm$ 1.06 | 38.94 $\pm$ 5.32 |
> > |  | 16 | **6.84 $\pm$ 0.70** | 24.92 $\pm$ 5.67 | 24.92| 8.68 $\pm$ 1.05 | 26.05 $\pm$ 5.63 |
> > |  | 20 |  **8.83$\pm$ 1.47** | 21.53 $\pm$ 7.71 | 21.53 | 9.09 $\pm$ 1.19 | 22.66 $\pm$ 4.70 |
> > |  | 25 |  **9.24$\pm$ 1.35** | 14.19 $\pm$ 5.11 | 14.19 | **9.91 $\pm$ 1.41** | 15.51 $\pm$ 3.32 |
> > | Auto-mpg | 4 | **2.99 $\pm$ 0.82** | 56.92 $\pm$ 13.00 | 56.92 | 4.83 $\pm$ 2.28 | 56.08 $\pm$ 6.53 |
> > |  | 6 | **3.83 $\pm$ 1.70** | 37.31 $\pm$ 14.10 | 37.31 | 5.64 $\pm$ 2.14 | 36.46 $\pm$ 9.47 |
> > |  | 8 | **6.14 $\pm$ 2.14** | 22.95 $\pm$ 19.88 | 22.95 | **6.30 $\pm$ 2.03** | 23.67 $\pm$ 9.82 |
> > |  | 13 | 7.42 $\pm$ 2.83 | 12.56 $\pm$ 6.83 | 12.56 | **6.68 $\pm$ 1.93** | 10.63 $\pm$ 3.54 |
> > | Housing | 9 | **6.25 $\pm$ 3.22** | 84.46 $\pm$ 11.67 | 84.46 | 8.56 $\pm$ 4.70 | 84.90 $\pm$ 8.22 |
> > |  | 12 | **7.40 $\pm$ 1.48** | 44.65 $\pm$ 8.69 | 44.65 | 9.76 $\pm$ 2.82 | 44.80 $\pm$ 8.62 |
> > |  | 16 | **8.35 $\pm$ 1.58** | 22.38 $\pm$ 8.90 | 22.38 | 10.52 $\pm$ 3.70 | 22.55 $\pm$ 6.50 |
> > |  | 20 | **9.59 $\pm$ 3.50** | 8.51 $\pm$ 6.82 | 8.51 | 11.16 $\pm$ 3.72 | 10.20 $\pm$ 4.14 |
> > | Concrete | 20 | **13.17 $\pm$ 4.91** | 69.42 $\pm$ 6.92 | 69.42 | 18.96 $\pm$ 5.71 | 71.17 $\pm$ 4.93 |
> > |  | 30 | **19.29 $\pm$ 3.85** | 44.08 $\pm$ 8.81 | 44.08 | 25.13 $\pm$ 4.57 | 48.20 $\pm$ 8.23 |
> > |  | 40 | **23.12 $\pm$ 4.59** | 31.50 $\pm$ 8.98 | 31.50 | 26.69 $\pm$ 4.83 | 33.74 $\pm$ 7.59 |
> > |  | 50 | **27.90 $\pm$ 4.31** | 19.76 $\pm$ 7.54 | 19.76 | 29.26 $\pm$ 4.56 | 21.36 $\pm$ 5.37 |
> > |  | 60 | **30.33 $\pm$ 4.89** | 12.82 $\pm$ 6.62 | 12.82 | **31.23 $\pm$ 4.65** | 14.90 $\pm$ 4.81 |

---

> > > ### Comment · Reviewer_J2H6 · 2023-08-20
> > >
> > > Thanks for clarifying the experimental setup. It helped me understand what you are doing. Now it looks more convincing than the experiments in the original submission.

---

### Comment · Area_Chair_aZ7n · 2023-08-18
**Reviewers, please respond to author's rebuttal.**

As a minimum, please acknowledge that you have read the rebuttal and whether it helps to change your rating, as the authors have tried to respond to your comments in the review. Thank you.

---

### Decision · Program_Chairs · 2023-09-21

**Decision:**

Accept (poster)

**Comment:**

This paper tackles the regression with rejection problem, which is of significant importance in the field. The paper solves the problem from a novel perspective and can be seen as a novel combination of several well-known techniques. This paper addresses clearly how it is related and different from related publications. The paper is technically sound and self-contained. Its claims are properly supported by theoretical demonstrations. The paper is clearly written and easy to follow. All reviewers agree that this is a good paper. Hence, I support acceptance of this paper.